# Cell type-specific network analysis in Diversity Outbred mice identifies genes potentially responsible for human bone mineral density GWAS associations

**Luke J Dillard[1], Gina Calabrese[1], Larry Mesner[1], Charles Farber[1,2]\***

[1]Department of Genome Sciences, University of Virginia, Charlottesville, United States; [2]Department of Biochemistry and Molecular Genetics, School of Medicine, University of Virginia, Charlottesville, United States

## eLife Assessment

This is an **important** study that provides **compelling** data from a diverse set of approaches from single cell transcriptome data and network analysis from genetically diverse mouse cells to identify novel driver genes underlying human GWAS associations. The authors present evidence that network analysis of scRNA-seq data from genetically diverse mouse bone-marrow derived stromal cells can be informative for identifying human BMD GWAS driver genes. Their approach should be broadly used and applicable to other GWAS studies.

**\*For correspondence:**
crf2s@virginia.edu

**Competing interest:** The authors declare that no competing interests exist.

**Abstract** Genome-wide association studies (GWASs) have identified many sources of genetic variation associated with bone mineral density (BMD), a clinical predictor of fracture risk and osteoporosis. Aside from the identification of causal genes, other difficult challenges to informing GWAS include characterizing the roles of predicted causal genes in disease and providing additional functional context, such as the cell-type predictions or biological pathways in which causal genes operate. Leveraging single-cell transcriptomics (scRNA-seq) can assist in informing BMD GWAS by linking disease-associated variants to genes and providing a cell-type context for which these causal genes drive disease. Here, we use large-scale scRNA-seq data from bone marrow-derived stromal cells cultured under osteogenic conditions (BMSC-OBs) from Diversity Outbred (DO) mice to generate cell type-specific networks and contextualize BMD GWAS-implicated genes. Using trajectories inferred from the scRNA-seq data that map cell state transitions, we identify networks enriched with genes that exhibit the most dynamic changes in expression across trajectories. We discover 21 network driver genes, which are likely to be causal for human BMD GWAS associations that colocalize with expression/splicing quantitative trait loci (eQTLs/sQTLs). These driver genes, including *Fgfrl1* and *Tpx2,* along with their associated networks, are predicted to be novel regulators of BMD via their roles in the differentiation of mesenchymal lineage cells. In this work, we showcase the use of single-cell transcriptomics from mouse bone-relevant cells to inform human BMD GWAS and prioritize genetic targets with potential causal roles in the development of osteoporosis.

## Introduction

Osteoporosis is a complex disease characterized by low bone mineral density (BMD), bone fragility, and an increased risk of fracture (*Lin and Lane, 2004*). BMD, a highly heritable trait, is one of the most important clinical predictors of osteoporotic fracture (*Peacock et al., 2002*; *Johnell et al.,*

*2005*). Increasing our understanding of the genetic basis of BMD is critical for the development of approaches for the treatment and prevention of osteoporosis. In recent years, genome-wide association studies (GWAS) have made great progress in unraveling BMD genetics by identifying over 1100 independent associations (*Morris et al., 2019*). Now the challenge lies in pinpointing causal genes, which is necessary for the translation of genetic findings into novel therapies.

A number of approaches exist to identify genes responsible for GWAS associations (*Cookson et al., 2009*; *Wen et al., 2017*; *Al-Barghouthi et al., 2022*; *Li and Ritchie, 2021*). Most rely on population-based '-omics' data (*Akiyama, 2021*), which are scarce for human bone, to connect associations to causal genes. Our group has used co-expression networks generated from mouse bone transcriptomic datasets to assist in the identification of genes likely responsible for BMD associations. One significant advantage of this approach is its ability to utilize the network connections of candidate genes to predict how these candidate genes may affect BMD. For example, we generated co-expression networks from bone tissue and primary osteoblasts in mouse genetic reference populations and identified multiple co-expression modules enriched with genes located in BMD associations (*Calabrese et al., 2017*; *Sabik et al., 2020*). We then cross-referenced genes in these modules with those regulated by colocalizing expression quantitative trait loci (eQTLs) from the Gene-Tissue Expression project (GTEx) (*GTEx Consortium, 2013*; *Aguet et al., 2020*) to identify 'high-priority' genes. Recently, we expanded our analyses to include directed networks generated via a Bayesian approach using cortical bone RNA-seq data from 192 Diversity Outbred (DO) mice. By combining key driver analysis and GTEx eQTL colocalization data, we identified 19 novel genes, such as *SERTAD4* and *GLT8D2*, which are likely causal for human BMD GWAS associations (*Al-Barghouthi et al., 2021*).

To date, our analyses have been reliant on networks generated from heterogeneous bulk transcriptomics (RNA-seq) datasets from mouse bone and primary bone cells. However, leveraging single-cell transcriptomics (scRNA-seq) data would offer the added benefit of resolving the transcriptomic profiles of discrete cell types. Additionally, using scRNA-seq data has the potential to provide context by predicting the specific cell types in which causal genes and their associated networks operate. In recent work, we demonstrated the utility of bone marrow-derived stromal cells cultured under osteogenic conditions (BMSC-OB) for the generation of population-scale scRNA-seq data on bone-relevant cell types (*Dillard et al., 2023*). The BMSC-OB model effectively enriches for mesenchymal lineage cells (e.g. mesenchymal progenitors, osteoblasts, osteocyte-like cells [Ocy]) that are highly relevant to the regulation of BMD.

In this work, our goal was to prioritize and contextualize genes implicated by BMD GWAS using an expanded large-scale (N=80) BMSC-OB scRNA-seq dataset on bone cell types. We accomplished this by first generating co-expression and Bayesian networks (*Al-Barghouthi et al., 2021*) for each BMSC-OB mesenchymal cell type. We subsequently prioritized networks based on their enrichment for genes exhibiting the most dynamic changes in expression across trajectories inferred from the scRNA-seq data, thus highlighting networks likely associated with the differentiation of BMSC-OBs. We then used these networks linked to osteogenic differentiation to prioritize genes with eQTLs and/ or splicing quantitative trait loci (sQTLs) which colocalize with BMD GWAS associations (*Al-Barghouthi et al., 2022*; *Abood et al., 2023*). In doing so, this analysis provides additional support for a role of these genes in the regulation of BMD and highlights their potential roles in differentiation of cell types essential to skeletal health.

## Results

### BMSC-OBs derived from DO mice yield diverse cell types that are enriched for mesenchymal lineage cells

We cultured BMSCs from a total of 80 DO mice, a genetically diverse outbred mouse population (*Bogue et al., 2015*; *Churchill et al., 2012*) (N=75 from the current study and N=5 from *Dillard et al., 2023*; N = 49 male and N = 31 females). We cultured BMSCs under osteogenic conditions and subsequently performed scRNA-seq, as described in *Dillard et al., 2023*. After stringent processing and quality control (Materials and methods), the dataset consisted of 21,831 expressed genes across 139,392 cells. We manually annotated 15 clusters ranging in size from 270 to 27,291 cells and identified cell types of the mesenchymal lineage, as well as various other cell types (*Figure 1A*, *Supplementary file 1a*, *Figure 1—figure supplement 1*).

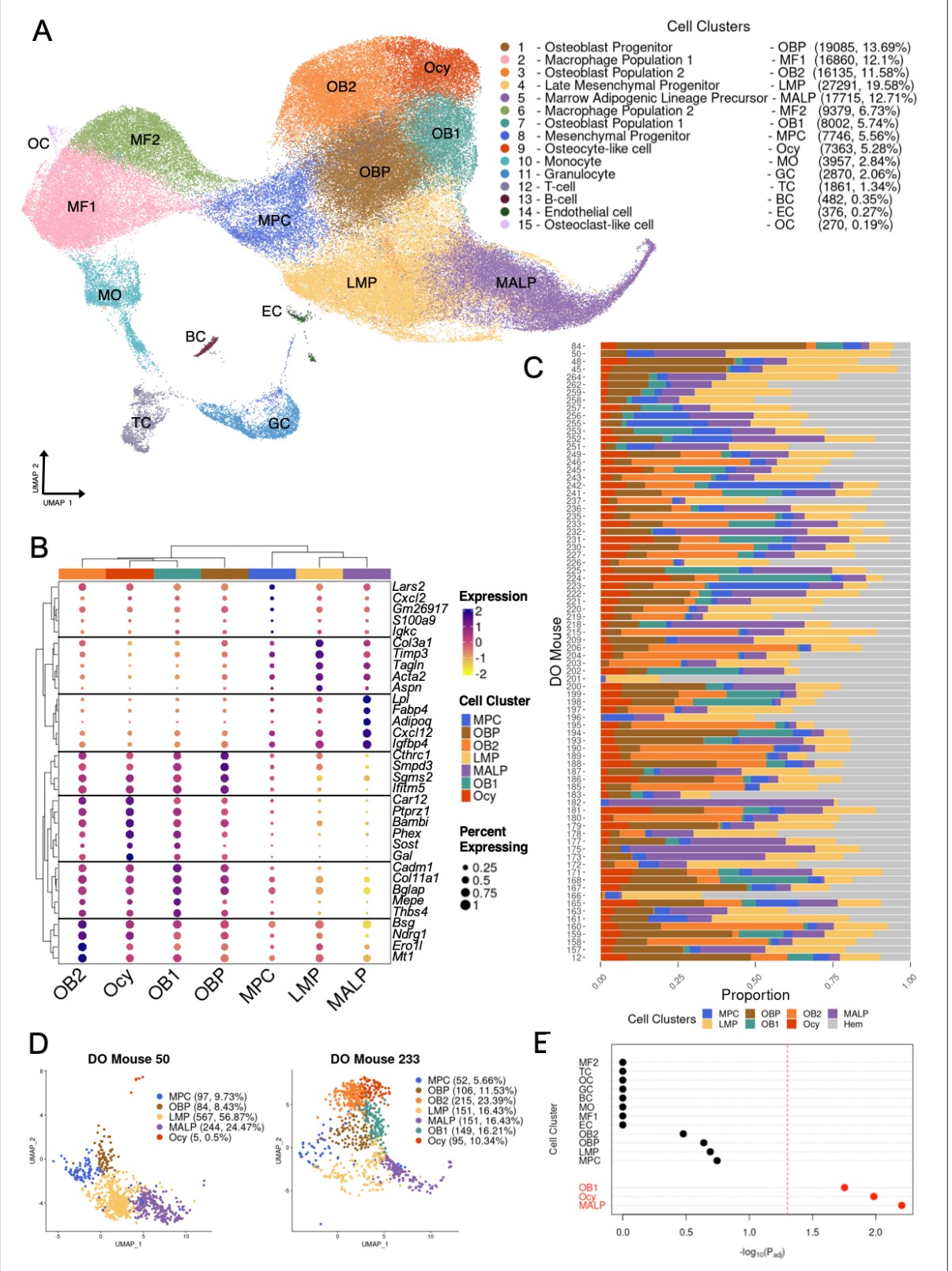

**Figure 1.** Analysis of single-cell RNA-seq (scRNA-seq) data for bone marrow-derived stromal cells cultured under osteogenic conditions (BMSC-OBs) derived from 80 Diversity Outbred (DO). (**A**) Uniform Manifold Approximation and Projection (UMAP) of 139,392 single cells (BMSC-OBs). Cell numbers and corresponding percentages for the fifteen (15) annotated cell clusters are listed in parenthesis to the right of the annotated cluster name. (**B**) Dot plot (*Marsh et al., 2023*) portraying representative and highly expressed genes for all annotated cell clusters. Dot color indicates the scaled gene

*Figure 1 continued on next page*

*Figure 1 continued*

expression while the size of the dot corresponds to the percentage of cells of a given cluster that express a given gene. (**C**) The raw proportional abundances of seven (7) mesenchymal cell clusters and one (1) cluster (Hem) representing the remaining cells (i.e. mainly hematopoietic immune cells) across all 80 DO mice. (**D**) UMAP plots for mesenchymal lineage cell clusters for DO mouse 50 and DO mouse 233. (**E**) CELLECT (CELL-type Expression-specific integration for Complex Traits) cell-type prioritization results displaying the Bonferroni adjusted p-values for the cell clusters. The OB1, Ocy, and marrow adipogenic lineage progenitor (MALP) cell clusters (red) were significantly enriched ($p_{adj}<0.05$, red dashed line) for BMD heritability ($p_{adj}$ = 0.018, 0.010, 0.006, respectively).

The online version of this article includes the following figure supplement(s) for figure 1:

**Figure supplement 1.** Plots displaying the distribution of the total number of cells from each mouse (N=80).

Based on our prior BMSC-OB scRNA-seq study (*Dillard et al., 2023*), we expected to identify a large proportion of mesenchymal cells and a smaller fraction of non-mesenchymal cell types. Consistent with this hypothesis, clusters associated with mesenchymal lineages accounted for 74.1% of all cells (*Figure 1A*). These included mesenchymal progenitor cells (MPCs), late mesenchymal progenitors (LMPs), osteoblast progenitors (OBPs), two mature osteoblast populations (OB1 and OB2), Ocy, and marrow adipogenic lineage progenitors (MALPs). The non-mesenchymal cell types observed included macrophages, monocytes, granulocytes, T-cells, B-cells, endothelial cells, and osteoclast-like cells (*Figure 1A*). With regard to the mesenchymal cell types, the only differences in cell clusters relative to our previous report (*Dillard et al., 2023*) were the presence of MPCs and two mature osteoblast clusters. Upon comparing the two distinct osteoblast clusters, OB1 and OB2 (*Figure 1A*), both clusters had ubiquitous expression of genes associated with mature osteoblasts (e.g. *Col1a1*, *Bglap*, *Sparc*, and *Ibsp*) (*Supplementary file 1a*) while many of the 'canonical' osteoblast markers were more highly expressed in OB1 compared to OB2 (*Supplementary file 1b*). Interestingly, MPCs did not have transcriptomic profiles similar to other MPCs previously identified by our group or others (*Dillard et al., 2023*; *Zhong et al., 2020*). All other mesenchymal cell types demonstrated specific expression of canonical marker genes (*Figure 1A and B*).

We next assessed the variability in cell-type frequencies across DO mice by quantifying the proportions of each annotated mesenchymal cell type. All other clusters, which mainly consisted of immune cells of hematopoietic origin, were aggregated into one group (Hem) for each mouse. We observed high variability in the raw proportional abundances of cell types derived from each mouse (*Figure 1C*, *Supplementary file 1c*). For example, the proportions of osteoblasts (OB1 and OB2) varied significantly among individual DO mice (*Figure 1D*). All mice used in the current experiment had been extensively phenotyped for a wide range of bone traits (microCT, histomorphometry, biomechanical bone properties, etc.) as part of a previous genetic analysis of bone strength (*Al-Barghouthi et al., 2021*). We correlated cell-type frequencies with bone traits; however, none of the cell-type proportions were strongly correlated with any bone trait (*Supplementary file 1d and e*).

## Mesenchymal lineage cells are enriched in BMD heritability

The primary goal of this work was to prioritize and contextualize genes implicated by BMD GWAS. As a first step toward this goal, we sought to determine which cell types were the most relevant to the genetics of BMD. Using the BMD GWAS and the BMSC-OB scRNA-seq data, we performed a CELLECT (*Timshel et al., 2020*) cell-type prioritization analysis to identify cell clusters enriched for BMD heritability. We observed that OB1, Ocy, and MALP cell clusters were significantly enriched ($p_{adj}<0.05$, red dashed line) for BMD heritability ($p_{adj}$=0.018, 0.010, 0.006, respectively) (*Figure 1E*, *Supplementary file 1f*). None of the non-mesenchymal cells identified were significant ($p_{adj}>0.05$) (*Figure 1E*). As a result, all downstream efforts focused solely on using data on mesenchymal cell types to inform GWAS.

## Generating mesenchymal cell type-specific Bayesian networks to inform BMD GWAS

We have previously shown that network-based approaches using bulk RNA-seq are powerful tools for the identification of putative causal genes from BMD GWAS data (*Calabrese et al., 2017*; *Sabik et al., 2020*; *Al-Barghouthi et al., 2021*). Here, our goal was to apply these same approaches using the BMSC-OB scRNA-seq data to prioritize and contextualize genes we previously identified as having a colocalizing eQTL (N=512) or sQTL (N=732) in a tissue from the GTEx project (*Al-Barghouthi et al.,*

*2022*; *Aguet et al., 2020*; *Abood et al., 2023*). Genes identified in each study (or both) yielded a list of high-priority target genes (N=1037). While GTEx does not currently contain data for bone tissue, eQTL can be shared across many tissues and may exert their effects in cell types resident to bone (*GTEx Consortium, 2017*). Therefore, utilizing our previous work, we hypothesized that generating cell type-specific networks would yield more biologically relevant representations of processes occurring within particular mesenchymal cell types. Additionally, by integrating GWAS, cell type-specific networks, and dynamic gene expression as a function of differentiation, we anticipated we would identify points of intervention in which genetic variation impacts genes involved in the differentiation process.

Our network analysis begins by partitioning genes into groups based on co-expression by applying iterative weighted gene co-expression network analysis (iterativeWGCNA) (*Greenfest-Allen et al., 2017*) to each mesenchymal cell type (Step 1, *Figure 2*). In total, 535 modules were identified from the BMSC-OB scRNA-seq data, and the number of modules identified for each mesenchymal cell cluster ranged from 26 to 153 with an average of 76 modules per cluster (*Supplementary file 1g and h*). We sought to infer causal relationships between genes in each cell type-specific co-expression module and subsequently identify networks involved in processes relevant to BMSC-OB differentiation. To this end, we generated Bayesian networks for each co-expression module, thus enabling us to model directed interactions between co-expressed genes based on conditional independence (*Al-Barghouthi et al., 2021*) (Step 2, *Figure 2*).

## Identifying putative drivers of mesenchymal cell differentiation

We hypothesized that many genes impacting BMD do so by influencing osteogenic differentiation or possibly bone marrow adipogenic differentiation of key mesenchymal cell types, as suggested by the CELLECT analysis above. Therefore, the next step of our network analysis was to identify cell type-specific Bayesian networks enriched for genes potentially driving mesenchymal differentiation (Step 3, *Figure 2*). To accomplish this, we first performed a pseudotime trajectory analysis to infer paths of differentiation in the mesenchymal lineage cells. We resolved three pseudotime trajectories (two osteogenic, one adipogenic) originating from the MPC cell cluster and ending in either Ocy, OB2, or MALP cell fates (*Figure 3A*). It is important to note that given the identification of multiple skeletal stem cells (*Chan et al., 2018*; *Mizuhashi et al., 2018*; *Debnath et al., 2018*; *Matsushita et al., 2020*), we do not view these trajectories as lineages, but rather 'differentiation paths' (progenitor to mature and/or terminally differentiated cells) that are likely traversed by different subsets of skeletal stem cells.

To identify genes likely impacting BMSC-OB differentiation, we used tradeSeq to identify genes that exhibit dynamic changes in expression along pseudotime (*Van den Berge et al., 2020*). Prior to performing the tradeSeq analysis, we parsed the pseudotime trajectories into regions that encompass cells associated with each cell type along their respective trajectories (*Figure 3B*). We defined multiple cell-type boundaries (nine in total) using pseudotime values, which represent points along a trajectory. The tradeSeq analysis was performed for each boundary (*Supplementary file 2a*). For example, trajectories bifurcate in the LMP cell cluster (*Figure 3A*); therefore, cells belonging to the LMP cluster can map to adipogenic and/or osteogenic trajectories depending on their placement along pseudotime. Between a cell-type boundary, cells spanning a specific cluster (e.g. LMP) and mapping to a specific trajectory (e.g. osteogenic trajectory) are used as input to analyze gene expression between the start and end points of the cell-type boundary (e.g. LMP_to_OBP). We analyzed gene expression within the established cell-type boundaries for all trajectories and identified genes that exhibit the most significant differences in expression between the start and end points of the cell-type boundaries. The total number of significant trajectory-specific tradeSeq genes ($p_{adj} < 0.05$) ranged from 87 to 1697 across the nine cell-type boundaries (*Supplementary file 2a and b–d*). The expression of representative marker genes for all cell types as a function of pseudotime was consistent with boundaries defined for each cell type (*Figure 3C*).

We sought to identify tradeSeq genes that may have an associated eQTL and hypothesized that eQTLs that perturb their expression would also impact the proportion of cells at different stages along the cell trajectories. We performed a cell type-specific eQTL analysis for each mesenchymal cell type from the 80 DO mice scRNA-seq data. We identified 563 genes (eGenes) regulated by a significant *cis*-eQTL in specific cell types of the BMSC-OB scRNA-seq data. Despite being significantly

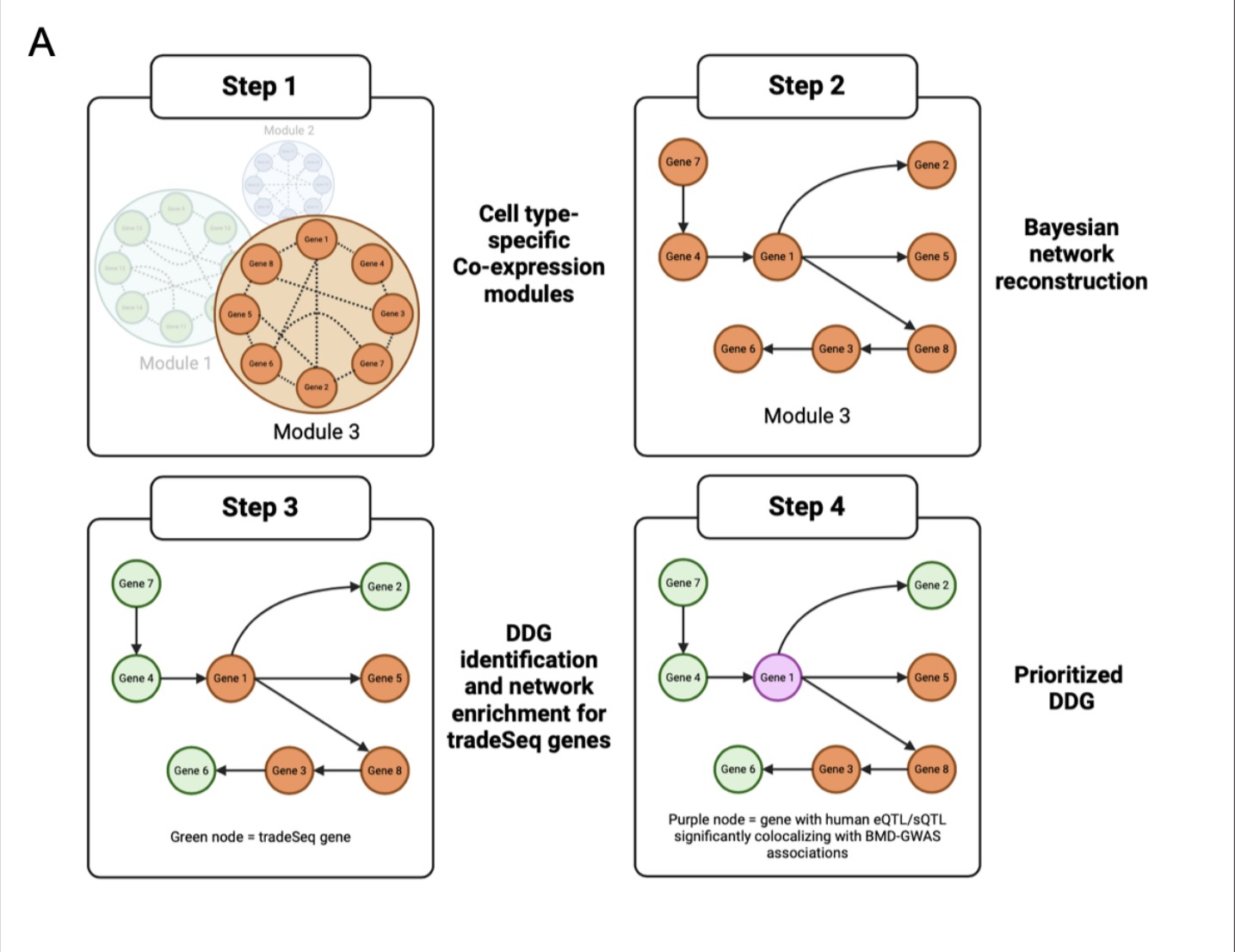

**Figure 2.** Overview of the network analysis pipeline. Step 1: For all seven (7) of the mesenchymal lineage cell clusters (mesenchymal progenitor cell [MPC], late mesenchymal progenitors [LMP], osteoblast progenitor [OBP], OB1, OB2, osteocyte-like cell [Ocy], marrow adipogenic lineage progenitor [MALP]), cell type-specific co-expression modules were generated using iterative Weighted Gene Co-expression Network Analysis (iterativeWGCNA). Step 2: Bayesian networks were learned to generate directed networks and model causal interactions between co-expressed genes. Step 3: Differentiation driver genes (DDGs) were identified by extracting subnetworks (i.e. large three-step neighborhood) for each gene in each cell type-specific Bayesian network and highlighting those subnetworks that were enriched ($p_{adj}$<0.05) for trajectory-specific tradeSeq genes for the cell-type boundary. Step 4: DDGs (and associated networks) were prioritized if the DDG was identified previously as an expression/splicing quantitative trait loci (eQTLs/sQTLs) that colocalized with BMD genome-wide association studies (GWAS) associations. Created with Biorender.com.

The online version of this article includes the following figure supplement(s) for figure 2:

**Figure supplement 1.** Scale-free topology and mean connectivity graphs for the cell type-specific iterative Weighted Gene Co-expression Network Analysis (iterativeWGCNA).

underpowered for this analysis due to our relatively smaller sample size (N=80), we identified two cell type-specific eGenes where the genotype responsible for the *cis*-eQTL effect was also associated with cell-type proportions. The first of these genes was Pyruvate Kinase, muscle (*Pkm*), which was identified as a significant global tradeSeq gene ($p_{adj}$=8.35 × $10^{-8}$; *Supplementary file 2e*) associated with the transition from LMPs to OBPs along an osteogenic trajectory (*Figure 4A*). Moreover, *Pkm* served as an eGene in the LMP cell cluster (LOD = 9.72; *Figure 4B*, *Supplementary file 2f*). Mice inheriting at least one PWK allele at this locus (N=15) demonstrated lower *Pkm* expression (*Figure 4C*) and a notable

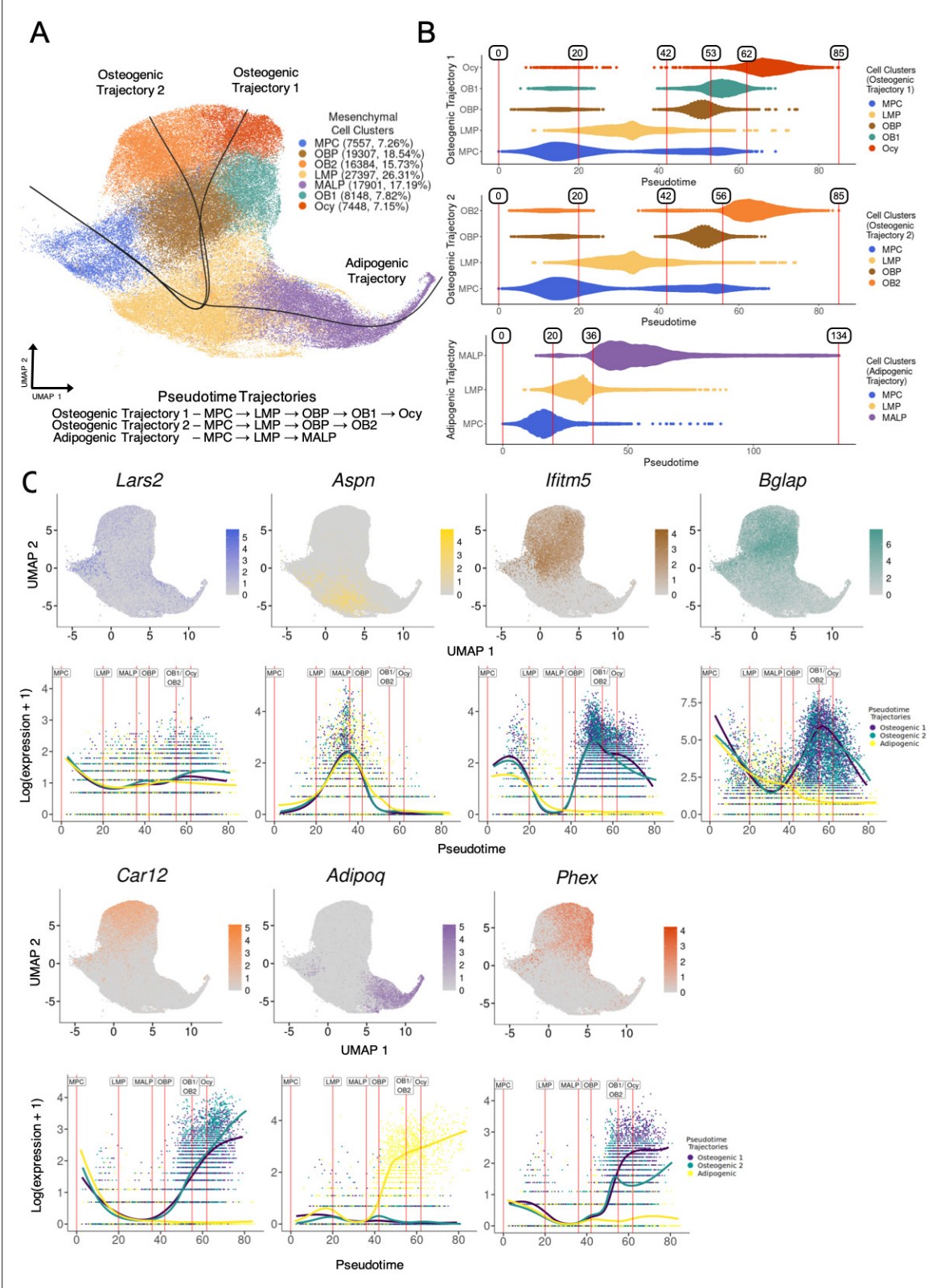

**Figure 3.** Pseudotime trajectory inference analysis and establishment of cell-type boundaries for tradeSeq analysis. (**A**) Three (3) trajectories (two adipogenic, one adipogenic) were inferred from the mesenchymal cell clusters of the bone marrow-derived stromal cells cultured under osteogenic condition (BMSC-OB) single-cell RNA-seq (scRNA-seq) data using Slingshot. All trajectories originate from the mesenchymal progenitor cell (MPC) and end in either osteogenic (osteocyte-like cells [Ocy], OB2) or adipogenic (marrow adipogenic lineage progenitor [MALP]) cell fates. (**B**) For each of the

*Figure 3 continued on next page*

*Figure 3 continued*

trajectories, cell-type boundaries were generated using pseudotime values along the trajectories, which encompass the majority of cells of a cell-type mapping to their respective trajectory. (C) Normalized gene expression of select genes associated with each cluster is represented in feature plots (*top*) and each gene plotted as a function of pseudotime (*bottom*) for all pseudotime trajectories (color corresponds to cell-type annotation observed throughout). Vertical lines (red) represent the cell-type (pseudotime) boundaries established for each cell type (label). The cell-type boundary for OB1 and OB2 is represented as one red line/label for visualization purposes.

reduction in mature osteoblasts (OB1) and Ocy proportions (p=0.030 and p=0.026, respectively), while LMP proportions were unaffected (*Figure 4D*, *Supplementary file 2g*).

Similarly, S100 calcium binding protein A1 (*S100a1*) was an OBP to OB1 transition tradeSeq gene ($p_{adj}$=0.023; *Figure 4A*, *Supplementary file 2e*) and an eGene in the OBP cell cluster (LOD = 10.12; *Figure 4B*, *Supplementary file 2f*). Mice inheriting at least one 129 allele at this locus (N=30) had higher *S100a1* expression, while the opposite was observed for mice inheriting NZO alleles (N=14) (*Figure 4C*). Additionally, mice inheriting at least one 129 allele showed a significant decrease in LMP proportion and increase in OB1 proportion (p=0.008 and p=0.016, respectively) (*Figure 4D*, *Supplementary file 2g*), while no significant differences were observed in cell-type proportions among mice inheriting NZO alleles at this locus (*Figure 4—figure supplement 1*, *Supplementary file 2g*).

## Identification of DDG

In order to discover BMSC-OB differentiation genes potentially responsible for BMD GWAS associations, the next step of our network analysis leveraged the trajectory-specific tradeSeq genes identified for each cell-type boundary (*Supplementary file 2b–d*) to identify differentiation driver genes (DDGs) (Step 3, *Figure 2*). We identified DDGs by extracting subnetworks (i.e. large three-step neighborhoods; see Methods) for each gene in each cell type-specific Bayesian network and identifying those subnetworks enriched ($p_{adj}$<0.05) for trajectory-specific tradeSeq genes for the cell-type boundary. The analysis identified 408 significant DDGs (*Supplementary file 2h–k*). We performed a PANTHER (*Thomas et al., 2022*) Gene Ontology (GO) analysis for the cell-type boundaries yielding a sufficient number of DDGs and found that DDGs for the osteogenic cell-type boundaries (LMP_to_OBP, OBP_to_OB1, OBP_to_OB2) were enriched for genes associated with the cell cycle (GO:0007049; N=23, 18, 23; p=1.12 × 10$^{-6}$, 1.29×10$^{-13}$, 1.0×10$^{-14}$, respectively) (*Supplementary file 3a–c*). The DDGs for the adipogenic cell-type boundary (LMP_to_MALP, MALP_to_end) were enriched for genes associated with extracellular matrix organization (GO:0030198; N=10; p=1.62 × 10$^{-7}$) and lipid metabolic processes (GO:0006629; N=25; p=1.83 × 10$^{-11}$), respectively (*Supplementary file 3d and e*). Across all 408 DDGs, 49 (12%) were identified in one or more cell-type boundaries as genes with a significant alteration (p<0.05) of whole-body BMD when knocked out/down in mice, as reported by the International Mouse Knockout Consortium (IMPC) (*Groza et al., 2023*; *Supplementary file 2i–k*).

We used our previously generated list of potentially causal BMD GWAS genes (N=1037) to subsequently prioritize the DDGs (Step 4, *Figure 2*). Of the 408 DDGs, 21 DDGs in one or more cell-type boundaries were genes that have BMD GWAS associations that colocalize with sQTL/eQTL (*Table 1*). The majority of these DDGs were identified in LMPs along both the osteogenic (LMP_to_OBP) and adipogenic (LMP_to_MALP) trajectories (N=10 and 6, respectively; *Supplementary file 2h*, *Supplementary file 3f*). The remaining DDGs were identified in OBPs along both osteoblast trajectories (OBP_to_OB1, OBP_to_OB2; N=1 and 3, respectively) and MALPs (MALP_to_end; N=6). Additionally, 3 of the 21 DDGs (*Tet1*, *Tpx2*, *Timp2*) are IMPC genes that exhibit a significant alteration of BMD (*Supplementary file 2h*, *Supplementary file 3f*).

## Network analysis predicts *Fgfrl1* and *Tpx2* as novel regulators of BMD

Here, we highlight two DDGs that putatively impact human BMD via their roles in LMP differentiation along either an adipogenic (*Fgfrl1*) or osteogenic (*Tpx2*) trajectory, which are genes with potential roles that have been minimally characterized in the context of human BMD. Based on our previous work (*Al-Barghouthi et al., 2022*), *Fgfrl1* (fibroblast growth factor receptor-like 1) was identified as a DDG with significant human BMD GWAS associations that also colocalized with eQTL identified in the cultured fibroblast GTEx tissue (RCP = 0.1611, *Table 1*). The *Fgfrl1* network was enriched for tradeSeq-identified genes (N=6 genes, $p_{adj}$ = 7.5 × 10$^{-3}$) for LMPs along an adipogenic trajectory (*Figure 5A*). An increase in the expression of all tradeSeq-identified genes for the *Fgfrl1* network was observed

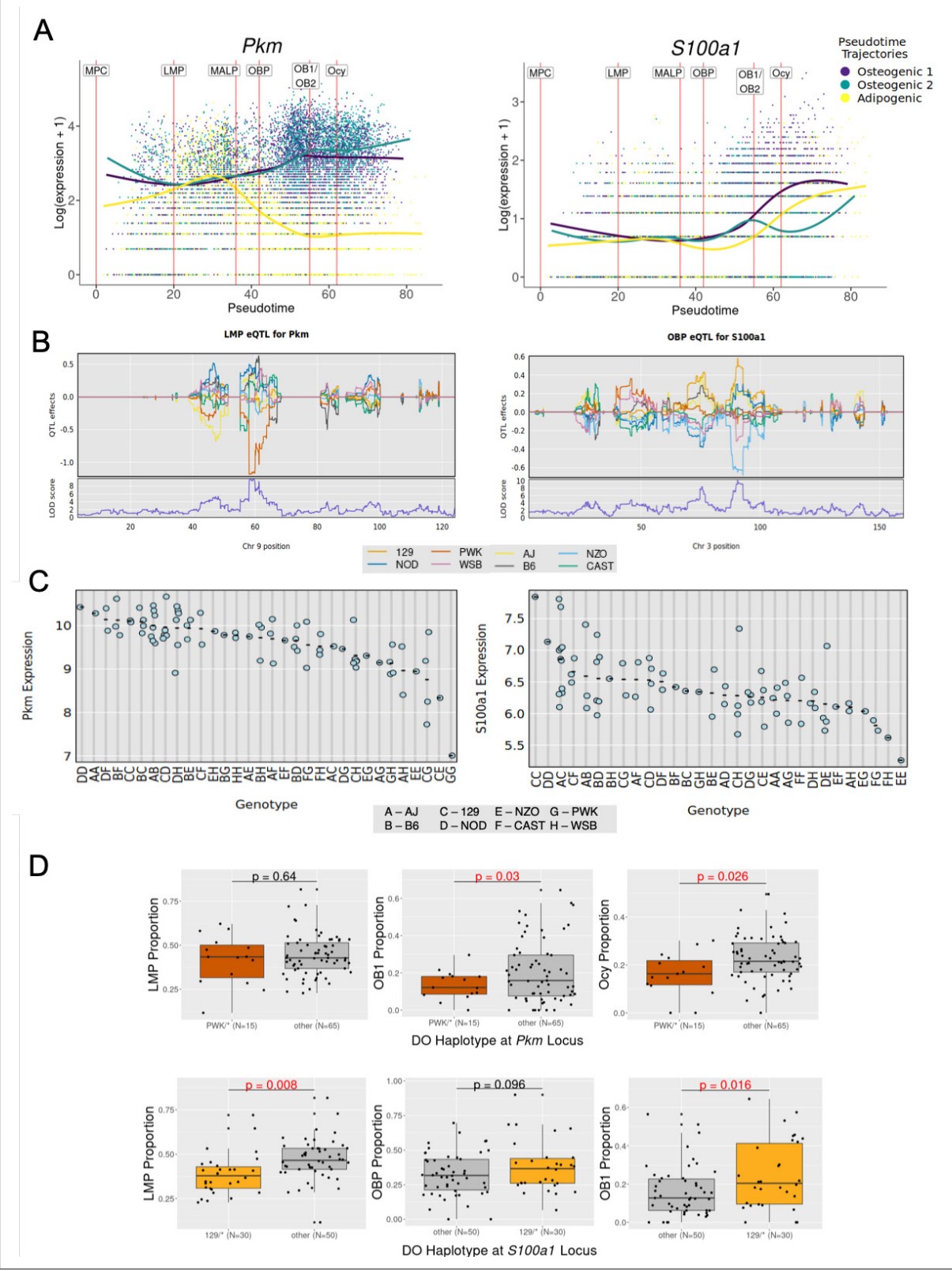

**Figure 4.** TradeSeq-identified genes associated with bone marrow-derived stromal cells cultured under osteogenic condition (BMSC-OB) differentiation exhibit expression quantitative trait locus (eQTL) effects. (**A**) *Pkm* was identified as a significant global tradeSeq-identified gene ($p_{adj}$ = 8.35 × 10$^{-8}$) for late mesenchymal progenitor (LMP) cells along an osteogenic trajectory (LMP_to_OBP) (*left*). *S100a1* was identified as a significant global tradeSeq-identified gene ($p_{adj}$=0.023) for OBP cells along osteogenic trajectory 1 (OBP_to_OB1) (*right*). (**B**) Plots indicating the cell type-specific eQTLs signal

*Figure 4 continued on next page*

*Figure 4 continued*

for both *Pkm* and *S100a1*. A negative eQTL effect on *Pkm* expression was observed in LMPs for Diversity Outbred (DO) mice with a PWK haplotype background at the *Pkm* locus (*left*). A positive eQTL effect on the expression of *S100a1* was observed in OBPs for DO mice with a 129 haplotype background at the *S100a1* locus, while a negative effect was observed for NZO mice (*right*). (**C**) The expression of *Pkm* and *S100a1* based on DO mouse (expression values transformed via variance stabilizing transformation [VST], as described in Methods). Genotype abbreviations correspond to DO haplotype background (legend) at the respective gene locus. Mice with at least one PWK allele (genotype abbreviation G) tend to have decreased expression of *Pkm* (*left*). Mice with at least one 129 allele (genotype abbreviation C) tend to have increased expression of *S100a1*, while NZO mice (genotype abbreviation E) have decreased expression (*right*). (**D**) PWK mice had a significant reduction in mature osteoblasts (OB1) and osteocyte-like cells (Ocy) proportions relative to other mice (p=0.030 and p=0.026, respectively; t-test), while LMP proportions were unaffected. Asterisks represent any of the other haplotype backgrounds. 129 mice showed a significant decrease in LMP proportion and increase in OB1 proportion (p=0.008 and p=0.016, respectively; t-test), but OBP proportions were unaffected. No significant effects on cell-type proportions were observed in NZO mice (*Figure 4—figure supplement 1*).

The online version of this article includes the following figure supplement(s) for figure 4:

**Figure supplement 1.** Tests of significance for cell-type proportions for NZO mice.

(*Figure 5B*, *Supplementary file 2d*). Importantly, the expression pattern for the tradeSeq-identified genes was consistent with the cell-type boundaries established for LMPs differentiating along the adipogenic trajectory toward the MALP cell state (*Figure 5B*). Furthermore, in the surrounding *Fgfrl1* network, two genes (*Plpp3* and *Cfap100*) have significant human BMD GWAS associations that also colocalized with sQTL in GTEx tissues, as reported in our previous study (*Abood et al., 2023*). In the *Fgfrl1* network, many other genes can be associated with adipocyte function (e.g. *Lpl*, *Plpp3*, *Igfbp4*) (*Enerbäck et al., 1992*; *Federico et al., 2018*; *Maridas et al., 2017*) and the maintenance of cilia (e.g. *Cfap100*, *St5* (*Denn2b*), *Mark1*) (*Sigg et al., 2017*; *Kumar et al., 2022*; *Fumoto et al., 2019*).

The other network we identified, the *Tpx2* network, was identified for LMPs along an osteogenic trajectory (*Figure 5C*). *Tpx2* (TPX2, microtubule-associated) is a DDG with significant human BMD GWAS associations that also colocalized with eQTL identified in the Testis GTEx tissue (RCP = 0.2031, *Table 1*). The network was enriched for tradeSeq-identified genes (N=9 genes, $p_{adj}$ = 5.7 × 10$^{-7}$) for LMPs differentiating along the osteogenic trajectory (*Figure 5C*). Furthermore, the expression of the tradeSeq-identified genes for the *Tpx2* network was consistent with the cell-type boundaries established for LMPs differentiating along the osteogenic trajectory toward the OBP cell state (*Figure 5D*; *Supplementary file 2b*). The expression of these genes increases as LMPs differentiate into OBPs and subsequently decreases upon reaching an OBP cell state. Additionally, *Tpx2* exhibited a significant alteration of BMD in both male and female mutant mice (genotype p-value = 1.03 × 10$^{-3}$) from IMPC (*Figure 5E*). In regard to the constituents of the *Tpx2* network, additional genes have been tested by the IMPC and result in a significant impact on BMD, such as *Ube2c*, *Top2a*, and *Papss1*. Many other genes in the *Tpx2* network can be associated with cellular division and proliferation, including four of the genes of the kinesin family (*Kif*) motor protein genes (*Miki et al., 2001*): *Kif4*, *Kif11*, *Kif15*, *Kif23*.

## Discussion

BMD GWAS has been successful at identifying thousands of SNPs associated with disease; however, the identification of causal genes and defining their functional role in disease remains challenging. The integration of '-omics' data, particularly transcriptomics, can assist in overcoming this challenge. Leveraging transcriptomics data has proven invaluable to informing GWAS, as demonstrated in studies that use these data to perform eQTL mapping, transcriptome-wide association studies, and co-expression/gene regulatory network reconstruction. GWAS associations can colocalize with predicted sources of genetic variation that perturb causal gene function or expression, thus providing a potential mechanism through which associations impact disease. While bulk RNA-seq data has been the foundation of such analyses, scRNA-seq data can provide valuable biological context by predicting the cell type in which causal genes are affected. To inform BMD GWAS, the generation of population-scale transcriptomics data at single-cell resolution in bone-relevant cell types can assist in the discovery of novel gene targets. Here, we perform scRNA-seq on 80 DO mice to generate single-cell transcriptomics data of mesenchymal cell types relevant to bone. Using these data, our goal was to prioritize putative causal genes and provide biological context in which these genes potentially influence disease, at cell type-specific resolution. Through our pseudotemporal gene expression and network analyses, we

**Table 1.** Prioritized differentiation driver genes (DDGs) that have bone mineral density (BMD) genome-wide association studies (GWAS) associations that colocalize with splicing/expression QTL (eQTL/sQTL) identified in a Genotype-Tissue Expression (GTEx) project tissue.

The tissue with the most significant colocalization (RCP and/or H4PP) is listed for each DGG (26 total, 21 distinct), as determined from *Al-Barghouthi et al., 2022*, and *Abood et al., 2023*, for eQTL and sQTL, respectively (*Al-Barghouthi et al., 2022*; *Abood et al., 2023*). RCP=Regional Colocalization Probability (GWAS and eQTL colocalization). H4P=H4 Posterior Probability (GWAS and sQTL colocalization).

| Trajectory | Cell-type boundary | DDG | GTEx Tissue with strongest eQTL colocalization (RCP) | GTEx Tissue with strongest sQTL colocalization (H4PP) | eGene identified from scRNA-seq of the 80 DO mice |
|---|---|---|---|---|---|
| 1 | LMP to OBP | Tet1 | Adipose (Visceral); 0.3191 | – | – |
| 1 | LMP to OBP | Tpx2 | Testis; 0.2031 | – | – |
| 1 | LMP to OBP | Cdk1 | – | Pituitary; 0.7795 | – |
| 1 | LMP to OBP | Ttyh3 | – | Liver; 0.9350 | – |
| 1 | LMP to OBP | Olfml3 | Artery (aorta); 0.8048 | – | – |
| 1 | LMP to OBP | Izumo4 | – | Brain (hypothalamus); 0.9182 | – |
| 1 | LMP to OBP | Sec24d | Nerve (tibial); 0.2677 | | – |
| 1 | LMP to OBP | Tmem263 | Adipose (subcutaneous); 0.5704 | Cultured cells (fibroblasts); 0.9716 | – |
| 1 | LMP to OBP | Lmf2 | – | Adrenal gland; 0.8181 | – |
| 1 | LMP to OBP | Tln2 | Esophagus (muscularis); 0.9697 | – | – |
| 1 | OBP to OB1 | Kremen1 | Heart (left ventricle); 0.8686 | – | – |
| 2 | OBP to OB2 | Kremen1 | Heart (left ventricle); 0.8686 | – | – |
| 2 | OBP to OB2 | Ebf1 | – | Testis; 0.8760 | – |
| 2 | OBP to OB2 | Lrp4 | Pancreas; 0.7943 | – | – |
| 3 | LMP to MALP | Ttyh3 | – | Liver; 0.9350 | – |
| 3 | LMP to MALP | Fgfrl1 | Cultured cells (fibroblasts); 0.1611 | – | – |
| 3 | LMP to MALP | Ebf1 | – | Testis; 0.8760 | – |
| 3 | LMP to MALP | Ppp1r12b | – | Nerve (tibial); 0.8807 | – |
| 3 | LMP to MALP | Rhoj | Cultured cells (fibroblasts); 0.352 | Breast; 0.7844 | – |
| 3 | LMP to MALP | Tln2 | Esophagus (muscularis); 0.9697 | – | – |
| 3 | MALP to end | Adh1 | – | Esophagus (gastroesophageal junction); 0.9999 | – |

*Table 1 continued on next page*

Table 1 continued

| Trajectory | Cell-type boundary | DDG | GTEx Tissue with strongest eQTL colocalization (RCP) | GTEx Tissue with strongest sQTL colocalization (H4PP) | eGene identified from scRNA-seq of the 80 DO mice |
| --- | --- | --- | --- | --- | --- |
| 3 | MALP to end | Fgfrl1 | Cultured cells (fibroblasts); 0.1611 | – | – |
| 3 | MALP to end | Adcy5 | – | Esophagus (gastroesophageal junction); 0.8456 | – |
| 3 | MALP to end | Cnn2 | – | Spleen; 0.7743 | – |
| 3 | MALP to end | Mxra8 | – | Pituitary; 0.7545 | – |
| 3 | MALP to end | Timp2 | – | Testis; 0.9429 | – |

identified 21 networks governed by predicted DDGs that have corresponding human BMD GWAS associations colocalizing with eQTL/sQTL in a GTEx tissue.

We demonstrate that the BMSC-OB model serves as an effective method to enrich for mesenchymal lineage cells, particularly bone-relevant cells. We characterized cells from 80 mice and identified both osteogenic and adipogenic cells derived from the mesenchymal lineage, such as two populations of osteoblasts (OB1 and OB2), Ocy, and MALPs. Our trajectory inference analysis identified three distinct trajectories in which MPCs give rise to both osteogenic and adipogenic cell types, thus portraying biologically relevant and known paths of differentiation of MPCs. Pseudotemporal gene expression was analyzed along each trajectory, in a cell type-specific fashion, to identify genes that were changing the most as a function of pseudotime (tradeSeq-identified genes). Subsequent *cis*-eQTL analysis indicated that the expression of some tradeSeq-identified genes was associated with the relative proportion of cell types. While instances such as these were rare, they illustrate that the potential consequence of genetic variation impacting the expression of tradeSeq-identified genes may impact differentiation and the abundances of certain cell types along a trajectory.

To inform BMD GWAS, we utilized the scRNA-seq data in a network analysis to contextualize causal genes (and their associated networks) by predicting the cell types through which these genes are likely acting. Toward this goal, we generated cell type-specific Bayesian networks from our BMSC-OB scRNA-seq data. Our approach was similar to our previous network analyses where bulk RNA-seq data was leveraged to identify genes with strong evidence of playing central roles in networks (*Calabrese et al., 2017*; *Sabik et al., 2020*; *Al-Barghouthi et al., 2021*). In contrast, here, we utilized scRNA-seq data to identify DDGs and prioritize networks based on the likelihood that they are involved in the differentiation of mesenchymal lineage cells (based on network connections enriched for tradeSeq-identified genes determined from inferred trajectories). Leveraging our previous work (*Al-Barghouthi et al., 2022*; *Abood et al., 2023*), we prioritized DDGs if they were genes with BMD GWAS associations colocalizing with human eQTL/sQTL in a GTEx tissue. Together, a gene being both a DDG and having BMD GWAS associations that colocalize with eQTL/sQTL is strong support of causality.

We identified 21 DDGs and associated networks, some of which have little to no known prior connection to bone. We contextualize these causal genes and their networks by not only providing cell-type predictions in which they likely operate, but also providing information regarding the biological processes they likely affect. For example, the *Tpx2* network was identified in LMPs differentiating along an osteogenic trajectory. *Tpx2* is a microtubule assembly factor that interacts with spindle microtubules during cellular division (*Zhang et al., 2017*). The expression of *Tpx2* and its regulation is associated with osteosarcoma, as well as other cancers (*Zhu et al., 2022*). In our previous study, *Tpx2* was identified as a gene that has BMD GWAS associations that colocalize with eQTL in the Testis GTEx tissue (*Al-Barghouthi et al., 2022*). While GTEx does not maintain bone tissue, eQTLs are shared across many tissues (*GTEx Consortium, 2017*); therefore, non-bone eQTLs may exert their effects in cell types associated with bone, such as LMPs, and evidence of a human eQTL effect indicates that genetic variation can modulate the expression of *Tpx2*. Additionally, when knocked out by IMPC,

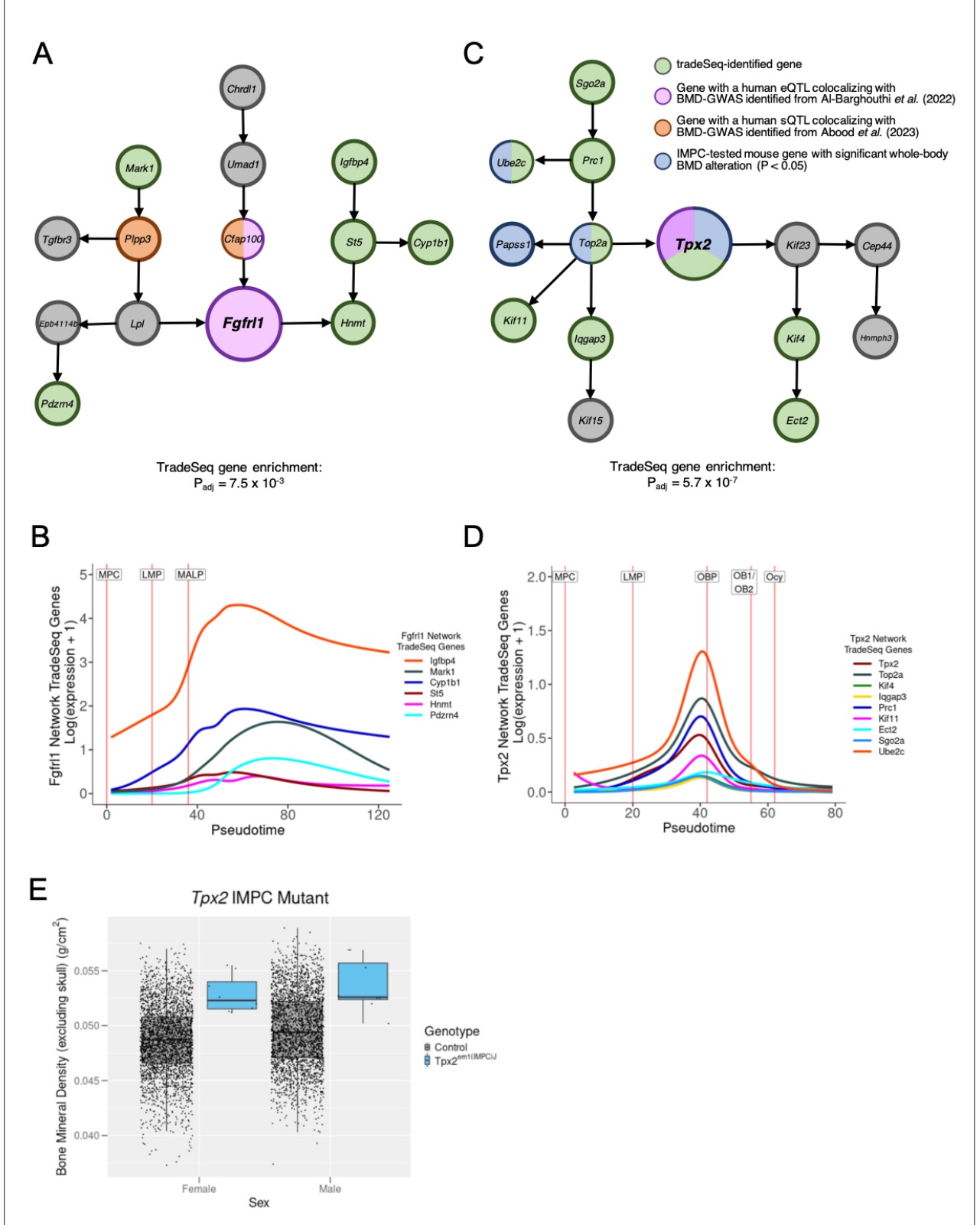

**Figure 5.** *Fgfrl1* and *Tpx2* are prioritized differentiation driver genes (DDGs) and putative drivers of mesenchymal differentiation. (**A**) *Fgfrl1* was identified as a DDG of a network for late mesenchymal progenitors (LMPs) differentiating along an adipogenic trajectory. The network is enriched ($p_{adj}$ = 7.5 × 10⁻³) for trajectory-specific tradeSeq-identified genes for the LMP_to_MALP cell-type boundary (*Hnmt, St5, Igfbp4, Cyp1b1, Pdzrn4, Mark1*). *Fgfrl1* was previously identified as a gene that has bone mineral density (BMD) genome-wide association studies (GWAS) associations that colocalize

*Figure 5 continued on next page*

*Figure 5 continued*

with an expression quantitative trait locus (eQTL) in the cultured fibroblast Genotype-Tissue Expression (GTEx) tissue. (**B**) An increase in the expression of tradeSeq-identified genes coincides with the LMP_to_MALP cell-type boundary in which they were identified as significant. (**C**) *Tpx2* was identified as a DDG of a network for LMPs differentiating along an osteogenic trajectory. The network is enriched ($p_{adj}$ = 5.7 × 10$^{-7}$) for tradeSeq-identified genes for the LMP_to_OBP cell-type boundary (*Tpx2, Top2a, Kif4, Iqgap3, Prc1, Kif11, Ect2, Sgo2a, Ube2c*). *Tpx2* is both a tradeSeq gene and previously identified as a gene that has BMD GWAS associations that colocalize with an eQTL in the Testis GTEx tissue. (**D**) An increase in the expression of tradeSeq-identified genes coincides with the LMP_to_OBP cell-type boundary in which they were identified as significant. (**E**) Box plot displaying whole-body BMD measurements (excluding skull) from the International Mouse Knockout Consortium (IMPC) for *Tpx2* mutant mice, which exhibited a significant increase in BMD (genotype p-value = 1.03 × 10$^{-3}$) in both male and female mice (N=8 (**M**) and 8 (**F**) mutants; N=2574 (**M**) and 2633 (**F**) controls).

*Tpx2* exhibited a significant increase in whole-body BMD in mice, thus providing strong support for *Tpx2* influencing the regulation of BMD in humans. In the surrounding gene neighborhood of the *Tpx2* network, other genes can be associated with cellular division as well, such as Topoisomerase 2A (*Top2a*) and the kinesin family (*Kif*) genes (*Miki et al., 2001*; *Uusküla-Reimand and Wilson, 2022*). Taken together, these results indicate a potential role of *Tpx2* as a mediator of BMD and genetic variation altering its expression could affect microtubule maintenance during the expansion of osteogenic cell populations.

Additionally, the *Fgfrl1* network was identified in LMPs differentiating along an adipogenic trajectory. Fibroblast growth factor receptor-like 1 (*Fgfrl1*) is presumed to function as a decoy receptor that interacts with FGF ligands necessary for FGF signaling (*Trueb, 2011*; *Steinberg et al., 2010*), and *Fgfrl1* expression is suggested to play a role in both adipogenic and osteogenic differentiation (*Kähkönen et al., 2018*). Our previous study also identified *Fgfrl1*, which has BMD GWAS associations that colocalize with eQTL in the cultured fibroblasts GTEx tissue (*Al-Barghouthi et al., 2021*). In the neighborhood of the *Fgfrl1* network, *Lpl, Plpp3, Igfbp4* have well-established roles in adipocyte function and metabolism (*Enerbäck et al., 1992*; *Federico et al., 2018*; *Maridas et al., 2017*); however, other genes can be associated with cilia, such as *Cfap100, St5* (*Denn2b*), *Mark1* (*Sigg et al., 2017*; *Kumar et al., 2022*; *Fumoto et al., 2019*). Interestingly, the maintenance and remodeling of cilia is essential to the differentiation of mesenchymal stem cells and pre-adipocytes (e.g. MALPs) while mature adipocytes lack cilia (*Hilgendorf, 2021*). Moreover, the inactivation of FGF signaling is associated with the length of primary cilia (*Neugebauer et al., 2009*). Thus, genetic variation altering the expression of expression of *Fgfrl1* may affect FGF signaling to impact the maintenance of cilia and adipogenic differentiation. Additionally, given the prioritization of MALPs in the CELLECT analysis and the well-established inverse relationship between marrow adiposity and BMD (*Fazeli et al., 2013*; *Veldhuis-Vlug and Rosen, 2018*), skewed balance of LMP differentiation toward marrow adipogenic cell fates may affect BMD. In summary, the *Fgfrl1* network harbors genes involved in adipogenic function, including cilia maintenance, which may contribute to LMP differentiation along an adipogenic trajectory. Together, these results indicate a potential role of *Fgfrl1* as a mediator of BMD via its role in adipogenic differentiation and maintenance of cilia.

Analyses performed here are not without limitations to consider. Our in vitro culturing approach and the preparation of single cells for scRNA-seq could be sources of technical variation in our study. Additionally, a pitfall of scRNA-seq is the sparsity of the resulting data, which yields an increased frequency of zero values for the expression of some genes in a proportion of cells, also known as 'drop-outs' (*Haque et al., 2017*). While statistical approaches can be employed to impute missing data, the accuracy of such methods and whether or not the resulting improvement in transcriptomic signal recovery is enough to warrant such intervention is contentious (*Cheng et al., 2023*; *Yu et al., 2021*). However, this issue may be partially offset given the larger scale of the scRNA-seq performed in this study and the average expression approach performed for network and eQTL analysis. Another limitation of this study is that read alignment of the scRNA-seq data did not account for DO founder genetic variation in RNA transcripts, which could affect read mapping and gene expression measurements. An additional limitation is that the BMSC-OB model does not capture osteoclasts, another cell type associated with bone tissue. Importantly, results from our CELLECT analysis indicate that BMD heritability was not enriched for genes whose expression was more specific to osteoclast-like cells; however, these cells likely represent immature osteoclasts, as mature multinucleated cells would be too large to be captured for sequencing. Lastly, while our study employed 80 DO mice, the issue of statistical power is still a limitation; however, we demonstrate that the BMSC-OB model is amenable

to high throughput and the inclusion of hundreds of mice, thus statistical power will be improved in future studies.

In summary, we showcase the use of large-scale scRNA-seq data to inform GWAS by performing a network analysis to contextualize BMD GWAS associations. Through the use of multiple single-cell analyses, we have expanded upon our understanding of the genetics of BMD. Our work exemplifies the power of single-cell transcriptomics from large populations of genetically diverse samples, and our network approach for data analysis may guide future studies to consider systems genetics strategies for the discovery of genetic determinants of disease.

## Methods

### Sample preparation and scRNA-seq

All animal procedures were conducted in compliance with the National Institutes of Health Guide for the Care and Use of Laboratory Animals. The protocol for studies involving DO mice (Protocol Number 3741) was reviewed and approved by the Institutional Animal Care and Use Committee (IACUC) at the University of Virginia. We prepared our samples in the same fashion as performed previously in *Dillard et al., 2023*. In brief, bone marrow was extracted from the femurs of initially 77 DO mice (The Jackson Laboratory, Strain: 009376). BMSCs were grown to confluence after 3 days of incubation in 48-well plates and then underwent in vitro osteoblast differentiation for 10 days with osteogenic differentiation media (alpha MEM, 10% FBS, 1% pen/strep, 1% GlutaMAX, 50 µg/µL ascorbic acid [Sigma, St. Louis, MO, USA], 10 nM β-glycerophosphate [Sigma], 10 nM dexamethasone [Sigma]). After differentiation, single cells were liberated from mineralizing cultures via incubations with 60 mM ethylenediaminetetraacetic acid pH 7.4 (EDTA [Thermo Fisher Scientific], made in DPBS), 8 mg/mL collagenase (Gibco) in HBSS/4 mM $CaCl_2$ (Fisher), and 0.25% trypsin-EDTA (Gibco). After single-cell isolation, cells from mice were pooled into groups containing cells from four to five mice total and concentrated to 800 cells/µL in PBS supplemented with 0.1% BSA (bovine serum albumin). Pooled single cells were prepared for sequencing using the 10× Chromium Controller (10× Genomics, Pleasanton, CA, USA) with the Single Cell 3' v2 reagent kit, according to the manufacturer's protocol. Libraries were sequenced on the NextSeq500 (Illumina, San Diego, CA, USA).

### scRNA-seq analysis pipeline

The data was subsequently processed using the 10× Genomics Cell Ranger toolkit (version 5.0.0) using the GRCm38 reference genome (*Church et al., 2009*). Using Seurat (*Hao et al., 2021*) (version 4.1.0), a combined Seurat object containing all cells was generated with the inclusion of features detected in at least three cells and cells with at least 200 features detected. We used Souporcell (*Heaton et al., 2020*) (version 2.0.0) to deconvolve the genotypes of all mice and to remove doublet cells. Cells were assigned to their associated DO mouse by making a pairwise comparison between allele calls made by the shared variants captured between Souporcell and GigaMUGA genotype arrays generated for all mice in the cohort, as previously performed in *Dillard et al., 2023*. Cells derived from two mice (176 and 244) were removed in some analyses due to poor genotyping of their respective Souporcell clusters, thus yielding a total of 75 DO mice from this study and 5 DO mice from our previous study (*Dillard et al., 2023*) for a total of 80 DO mouse biological replicates. We filtered out cells with more than 6200 reads and less than 400 reads, as well as those cells with more than 10% mitochondrial reads. Further, cells were removed if they expressed greater than 20% *Rpl* and 15% *Rps* reads, which equates to cells approximately exceeding the 98th percentile. After filtering, 139,392 cells remained, and the resulting object underwent standard normalization, scaling, and the top 3000 features were modeled from a variance stabilizing transformation (VST) using Seurat. Cell-cycle markers based on *Tirosh et al., 2016*, were regressed out using the 'CellCycleScoring' and scaling functions. For subsequent dimensionality reduction, 15 principal components (PCs) were summarized. Then, a kNN (k = 20) graph was created and the Louvain algorithm was used to cluster cells at a resolution of 0.5. Annotation of cell-type clusters was performed manually based on differential gene expression analysis using the Seurat 'FindAllMarkers' function (*Supplementary file 1a*).

For subsequent WGCNA and eQTL mapping, transcriptomic profiles for each cell-type cluster were generated for each sample using a mean expression approach, as performed similarly by others (*Neavin et al., 2021*; *van der Wijst et al., 2018*). For each sample contributing at least five cells to a

given cluster, unnormalized unique molecular identifier (UMI) counts of gene expression for all cells in the cluster for the sample were averaged and then rounded to the nearest hundredth decimal place. A total of 80, 80, 77, 67, 50, 76, 80 mice contributed enough cells to the MPC, LMP, OBP, OB1, OB2, Ocy, and MALP cell-type clusters, respectively. Genes with non-zero expression values in fewer than 15 samples were removed. A total of 11,971, 15,162, 14,857, 13,674, 13,825, 14,136, and 14,534 genes remained for the MPC, LMP, OBP, OB1, OB2, Ocy, and MALP clusters, respectively. Samples were normalized by computing CPMs (counts per million) without log transformation for each gene using edgeR (*Robinson et al., 2010*) (version 4.0.7), then transformed via VST using DESeq2 (*Love et al., 2014*) (version 1.42.0), and quantile normalized using preprocessCore (version 1.60.2).

## Trajectory and tradeSeq analysis

Trajectory inference analysis was performed using Slingshot (*Street et al., 2018*) (version 1.8.0) on the mesenchymal lineage cell clusters (seven total) of the BMSC-OB scRNA-seq data. The starting cluster was set as the MPC cluster upon the removal of a small outlier population of cells. Trajectories were inferred using 15 PCs. TradeSeq (*Van den Berge et al., 2020*) (version 1.4.0) was used to analyze gene expression along the trajectories by fitting a negative binomial generalized additive model (NB-GAM) to each gene using the 'fitGAM' function with nknots = 10, which was determined by using the 'eval-uateK' function. Prior to performing the tradeSeq analysis, the scRNA-seq data was downsampled to reduce the size of the dataset to approximately 10,000 cells (sampled at random across all seven clusters).

All cell-type boundaries were established to encompass, on average, 78% of cells of a cell cluster (*Supplementary file 2a*). To identify genes significantly changing between boundaries in a trajectory-specific fashion, we first performed tradeSeq to compare gene expression within each trajectory (two osteogenic, one adipogenic) to highlight genes with a significant difference in expression between boundaries using the 'startVsEndTest' function (*Supplementary file 2a–d*). Next, we performed a global test with tradeSeq to compare gene expression between trajectories in order to highlight genes exhibiting a significant difference in expression using the 'startVsEndTest' function (*Supplementary file 2a*, *Supplementary file 2e*). All tests were performed with the $\log_2$ fold change threshold (l2fc)=0.5. For all global and trajectory-specific tests, the p-values associated with each gene were adjusted to control the false discovery rate using the 'p.adjust' function from the stats (version 4.2.1) R package, and genes were filtered to include those with a $p_{adj}$<0.05.

## CELLECT analysis

CELLECT (*Timshel et al., 2020*) (CELL-type Expression-specific integration for Complex Traits) (version 1.1.0) was used to identify likely etiologic cell types underlying complex traits of both the BMSC-OBs scRNA-seq data (*Figure 1E*, *Supplementary file 1f*). CELLECT p-values were adjusted using the Bonferroni correction. CELLECT quantifies the association between the GWAS signal and cell-type expression specificity using the S-LDSC genetic prioritization model (*Finucane et al., 2015*). Summary statistics from the UK Biobank eBMD and Fracture GWAS (Data Release 2018) and cell-type annotations from each scRNA-seq data set were used as input. Cell-type expression specificities were estimated using CELLEX (*Timshel et al., 2020*) (CELL-type EXpression-specificity) (version 1.2.1) (*Supplementary file 3g*).

## WGCNA

Cell type-specific mean expression matrices (as obtained above) were used as input to generate signed co-expression network modules (*Supplementary file 1g and h*). IterativeWGCNA (*Greenfest-Allen et al., 2017*) (version 1.1.6) was used from a Singularity container built from a Docker hub image (*Cartailler, 2022*). A soft threshold (power) of 14, which exceeded an $R^2$ threshold of 0.85 for all cell-type clusters, was selected for module construction (*Figure 2—figure supplement 1*). Modules were generated using iterativeWGCNA with default parameters for the 'blockwiseModules' function, a minimum module size of 20 genes, minCoreKME = 0.7, and minKMEtoStay = 0.5.

## Bayesian network learning

Bayesian networks were learned from each of the cell type-specific modules of co-expressed genes with the bnlearn (version 4.8.3). Gene expression matrices containing the genes for each module

were used as input to the 'mmhc' function which employs the Max-Min Hill Climbing (MMHC) algorithm (*Tsamardinos et al., 2006*) to learn the underlying structure of the Bayesian network. From the generated networks, igraph (version 1.6.0) was used to resolve three-step neighborhoods (*Porter and Smith, 2010*). Nodes (genes) that were unconnected to a neighborhood or connected to only one neighbor were removed. Neighborhoods were filtered to include those with a size greater than 1 standard deviation from the mean across all neighborhoods generated for the network.

DDGs are genes that yield large three-step neighborhoods that are enriched ($p_{adj}<0.05$) with tradeSeq-identified genes for a given cell-type boundary. We calculated whether each neighborhood contained more tradeSeq-identified genes (for the neighborhoods' associated cell-type boundary) than would be expected by chance using the hypergeometric distribution ('phyper' function) from the stats (version 4.2.1) R package. The arguments were as follows: q: (number of neighbors in a neighborhood that are also tradeSeq-identified genes for a given cell-type boundary) – 1; m: total number of tradeSeq-identified genes for a given cell-type boundary; n: (total number of identified neighborhoods) – m; k: neighborhood size (total number of neighbors); lower.tail=false. p-Values were adjusted to control the false discovery rate using the 'p.adjust' function from the stats (version 4.2.1) R package. These pruning steps resulted in a total of 408 DDGs and associated networks for all cell types (*Supplementary file 2h–k*).

## DO eQTL mapping

Prior to performing the eQTL analysis, DNA was extracted from the tails of the 80 DO mice, using the PureLink Genomic DNA Mini Kit (Invitrogen) and genotyped using the GigaMUGA array by Neogen Genomics (GeneSeek; Lincoln, NE, USA). Processing and quality control of genotype data, including calculation of genotype/allele probabilities, was performed as previously described in *Al-Barghouthi et al., 2021*. Cell type-specific mean expression matrices (as obtained above) for mesenchymal lineage clusters were used as input for the eQTL mapping, which was performed using a linear mixed model via the 'scan1' function from the qtl2 (*Broman et al., 2019*) (version 0.30) R package with allowances for the following covariates: sex, age at sacrifice (in days), weight, length, and DO mouse generation. To identify significant eQTL, we calculated an LOD (logarithm of the odds) threshold; for each cell-type cluster, we chose 50 genes at random and then permuted them 1000 times using the 'scan1perm' function from qtl2. We established the LOD threshold of 9.68 and 9.49 for the autosomal chromosomes and X chromosome, respectively, by taking the average of the median LOD across each cell type. A total of 563 eQTLs exceeded the LOD thresholds and were no more than 1 Mbp from the transcription start site of the associated eGene (*Supplementary file 2f*).

## Cell-type proportion analysis

To account for technical sources of variation often retained in scRNA-seq, cell-type proportions were transformed using the arcsin (asin) square root transformation from the speckle (*Phipson et al., 2022*) R package (version 0.0.3). Tests of statistical significance were performed using the propeller t-test and ANOVA functions with default parameters. The sex of the mice and the batch each mouse was associated with for sequencing were modeled as covariates. Transformed values were used as input for computing tests of statistical differences of cell-type proportions between mice, as well as correlation to phenotypic traits (*Supplementary file 1c–e*).

## Acknowledgements

Research reported in this publication was supported in part by the National Institute of Arthritis and Musculoskeletal and Skin Diseases of the National Institutes of Health under award numbers R01AR68345, R01AR082880, and R01AR077992 to CRF.

## Additional information

### Funding

| Funder | Grant reference number | Author |
|---|---|---|
| National Institute of Arthritis and Musculoskeletal and Skin Diseases | R01AR68345 | Charles Farber |
| National Institute of Arthritis and Musculoskeletal and Skin Diseases | R01AR082880 | Charles Farber |
| National Institute of Arthritis and Musculoskeletal and Skin Diseases | R01AR077992 | Charles Farber |

The funders had no role in study design, data collection and interpretation, or the decision to submit the work for publication.

### Author contributions

Luke J Dillard, Conceptualization, Data curation, Formal analysis, Methodology, Visualization, Writing – original draft, Writing – review and editing; Gina Calabrese, Larry Mesner, Visualization; Charles Farber, Methodology, Supervision, Funding acquisition, Writing – review and editing

### Author ORCIDs

Luke J Dillard ⓘ https://orcid.org/0000-0001-8293-0479
Charles Farber ⓘ https://orcid.org/0000-0002-6748-4711

### Ethics

All animal procedures were conducted in compliance with the National Institutes of Health Guide for the Care and Use of Laboratory Animals. The protocol for studies involving Diversity Outbred mice (Protocol Number 3741) was reviewed and approved by the Institutional Animal Care and Use Committee (IACUC) at the University of Virginia.

Reviewer #1 (Public review): https://doi.org/10.7554/eLife.100832.3.sa1
Reviewer #2 (Public review): https://doi.org/10.7554/eLife.100832.3.sa2
Author response https://doi.org/10.7554/eLife.100832.3.sa3

---

## Additional files

### Supplementary files

Supplementary file 1. Supplementary file 1a-h. (a) Differentially expressed genes (DEGs) for all clusters of the bone marrow-derived stromal cells cultured under osteogenic condition (BMSC-OB) scRNA-seq cell clusters. DEGs were calculated on all clusters of the BMSC-OB scRNA-seq data using the FindAllMarkers function from the Seurat R package. (b) DEGs between the OB1 and OB2 clusters of the scRNA-seq data. DEGs were calculated using the FindMarkers function from the Seurat R package. Positive values for average $\log_2$ fold change (avg_log2FC) indicate that a gene is more highly expressed in OB1. (c) BMSC-OB cell-type proportion analysis for the 80 Diversity Outbred (DO) mice. The raw proportions (top) and asin-transformed proportions (bottom) of each of the BMSC-OB cell types were calculated from the total number of cells contributed by each mouse using the Propeller R package. All non-mesenchymal lineage cell types (i.e. hematopoietic lineage cells) are aggregated as a group (Hem) for each mouse. (d) Correlation of cell proportions to bone trait metrics captured from the 80 DO mice. Raw (top) and transformed (bottom) cell-type proportions were correlated using Pearson and Spearman to bone trait metrics (55 total) captured on all mice from the 80 DO mice. (e) Bone trait abbreviations and units of measurement. (f) CELLECT cell-type prioritization table. Beta is a regression effect size estimate for given annotation. Beta SE is the standard error for the regression coefficient. The p-value is the one-

sided test (beta>0) association between bone mineral density (BMD) genome-wide association study (GWAS) signal heritability and each annotated cell type. p-Values were adjusted using the Bonferroni correction method. MALP=marrow adipogenic lineage precursors; Ocy=osteocyte-like cell; OB1=osteoblast population 1; MPC=mesenchymal progenitor cell; LMP=late mesenchymal progenitor; OBP=osteoblast progenitor; OB2=osteoblast population 2; EC=endothelial cell; MF1=macrophage population 1; MO=monocyte; BC=B-cell; GC=granulocyte; OC=osteoclast-like cell; TC=T-cell; MF2=macrophage population 2. (g) Summary of results from the iterative Weighted Gene Co-expression Network Analysis (iterativeWGCNA). A total of 535 co-expression modules were generated using the mesenchymal lineage cell clusters (7 total) of the BMSC-OB scRNA-seq data, yielding an average of 76 modules per cell cluster. A total of 8810 Bayesian networks were generated from the co-expression modules. (h) Genes within each module generated from the iterativeWGCNA.

Supplementary file 2. Supplementary file 2a-k. (a) Summary of tradeSeq-identified genes. For each cell-type (pseudotime) boundary associated with a specific trajectory (9 total), a global and trajectory-specific test was performed using the startVsEndTest function from the tradeSeq R Package. The number of genes identified for each test and for each boundary is displayed, as well as the number of tradeSeq-identified genes that were also identified as eGenes from the expression quantitative trait locus (eQTL) mapping of the 80 Diversity Outbred (DO) mice (73 total). (b) TradeSeq-identified genes from the trajectory-specific analysis for osteogenic trajectory 1. All significant trajectory-specific tradeSeq-identified genes ($p_{adj}$≤0.05) across all cell-type boundaries (5 total; MPC, LMP, OBP, OB1, Ocy) associated with osteogenic trajectory 1. Associated eQTL information is also displayed for the gene if it was an eGene identified in the cell type from the cell type-specific eQTL analysis (if 'NA' is present, the gene was not identified as an eGene). (c) TradeSeq-identified genes from the trajectory-specific analysis for osteogenic trajectory 2. All significant trajectory-specific tradeSeq-identified genes ($p_{adj}$≤0.05) across all cell-type boundaries (2 total; OBP, OB2) associated with osteogenic trajectory 2. Associated eQTL information is also displayed for the gene if it was an eGene identified in the cell type from the cell type-specific eQTL analysis (if 'NA' is present, the gene was not identified as an eGene). (d) TradeSeq-identified genes from the trajectory-specific analysis for the adipogenic trajectory. All significant trajectory-specific tradeSeq-identified genes ($p_{adj}$≤0.05) across all cell-type boundaries (2 total; LMP, MALP) associated with the adipogenic trajectory. Associated eQTL information is also displayed for the gene if it was an eGene identified in the cell type from the cell type-specific eQTL analysis (if 'NA' is present, the gene was not identified as an eGene). (e) TradeSeq-identified genes from the global analysis. All significant global tradeSeq-identified genes ($p_{adj}$≤0.05) across all cell type (pseudotime) boundaries (9 total). Associated eQTL information is also displayed for the gene if it was an eGene identified in the cell type from the cell type-specific eQTL analysis (if 'NA' is present, the gene was not identified as an eGene). (f) Results from the cell type-specific eQTL analysis on the mesenchymal lineage cell types identified in the scRNA-seq data from the 80 DO mice. All significant eQTLs (LOD>9.68 for autosomal chromosomes; LOD>9.49 for X chromosome) and eGenes for the mesenchymal cell clusters (563 total). Chr=chromosome of eQTL, Pos=eQTL peak position, LOD=logarithm of the odds score, ci (low/hi)=LOD support intervals, Start=start position of gene (GRCm38), End=end position of gene (GRCm38), dist_start=distance of eQTL to start. (g) Results from tests of significance for cell-type proportions. Tests of significance on the transformed cell-type proportions were performed using the Propeller R package, and nominal p-values are reported. Sample batch (pool containing cells from mice in preparation for scRNA-seq) and sex were modeled as covariates. For the Pkm example, a t-test was performed for all mice with at least one PWK haplotype background (PWK/*; asterisk meaning any DO haplotype) at a Pkm locus (n=15) against all remaining mice (n=65). For the S100a1 example, a one-way ANOVA was performed on four groups: mice with at least one 129/* haplotype background (n=26) or NZO/* (n=10), heterozygous for both (129/NOZ, n=4), or any other DO haplotype background at the locus (n=40). Additionally, t-tests were performed on the 129/* and NZO/* haplotype background individually against all other mice aggregated as a group. (h) Summary of differentiation driver gene (DDG) network analysis. The number of DDGs and associated networks that were enriched ($p_{adj}$≤0.05) with more genes in the trajectory-specific tradeSeq genes for each cell-type boundary (408 total). The number of DDGs that had a corresponding human homolog with a human BMD GWAS association (that colocalizes with expression and/or splicing quantitative trait loci [eQTLs/sQTLs] from the Genotype-Tissue Expression (GTEx) Project) is also displayed (26 total, 21 distinct). Three of the DDGs were also tested by the International Mouse Knockout Consortium (IMPC) and had a significant BMD phenotype when knocked out. (i) All significant DDG network analysis for osteogenic trajectory 1 (178 total). The enrichment of each DDG Bayesian network for tradeSeq-identified genes (identified

for each cell-type boundary along osteogenic lineage 1) is displayed as nominal and adjusted p-values, as well as the co-expression module in which the DDG was identified. The data can be filtered to highlight DDGs that are: a tradeSeq-identified gene for the cell boundary, a gene that was identified by *Al-Barghouthi et al., 2022*, as having eQTL that also colocalizes with BMD GWAS associations, a gene that was identified by *Abood et al., 2023*, as having sQTL that also colocalizes with BMD GWAS associations, a gene that was tested by the IMPC and had a significant effect on BMD when knocked out, or gene that was identified here as an eGene in the cell type-specific eQTL analysis using the 80 DO scRNA-seq data. Other columns of the data contain information at the network level: Number_neighbors=number of nodes (genes) in Bayesian network, Number_tradeSeq_gene_neighbors=number of genes that were also tradeSeq-identified genes for the cell-type boundary, All network neighbors=all genes in network, Neighbors_eQTL_Al_Barghouthi=all genes in the network that were identified from *Al-Barghouthi et al., 2022*, Neighbors_sQTL_Abood=all genes in the network that were identified from *Abood et al., 2023*, Neighbors_IMPC_BMD_gene=all genes in the network that were tested by the IMPC and had a significant effect on BMD when knocked out, Neighbors_DO_eGene=all genes in the network that was identified here as an eGene in the cell type-specific eQTL analysis using the 80 DO scRNA-seq data, Neighbors_tradeSeq_gene_for_boundary=all genes that were also tradeSeq-identified genes for the cell-type boundary. (j) All significant DDG network analysis for osteogenic trajectory 2 (55 total). The enrichment of each DDG Bayesian network for tradeSeq-identified genes (identified for cell-type boundary along osteogenic lineage 2) is displayed as nominal and adjusted p-values, as well as the co-expression module in which the DDG was identified. The data can be filtered to highlight DDGs that are: a tradeSeq-identified gene for the cell boundary, a gene that was identified by *Al-Barghouthi et al., 2022*, as having eQTL that also colocalizes with BMD GWAS associations, a gene that was identified by *Abood et al., 2023*, as having sQTL that also colocalizes with BMD GWAS associations, a gene that was tested by the IMPC and had a significant effect on BMD when knocked out, or gene that was identified here as an eGene in the cell type-specific eQTL analysis using the 80 DO scRNA-seq data. Other columns of the data contain information at the network level: Number_neighbors=number of nodes (genes) in Bayesian network, Number_tradeSeq_gene_neighbors=number of genes that were also tradeSeq-identified genes for the cell-type boundary, All network neighbors=all genes in network, Neighbors_eQTL_Al_Barghouthi=all genes in the network that were identified from *Al-Barghouthi et al., 2022*, Neighbors_sQTL_Abood=all genes in the network that were identified from *Abood et al., 2023*, Neighbors_IMPC_BMD_gene=all genes in the network that were tested by the IMPC and had a significant effect on BMD when knocked out, Neighbors_DO_eGene=all genes in the network that was identified here as an eGene in the cell type-specific eQTL analysis using the 80 DO scRNA-seq data, Neighbors_tradeSeq_gene_for_boundary=all genes that were also tradeSeq-identified genes for the cell-type boundary. (k) All significant DDG network analysis for the adipogenic trajectory (175 total). The enrichment of each DDG Bayesian network for tradeSeq-identified genes (identified for cell-type boundary along the adipogenic trajectory) is displayed as nominal and adjusted p-values, as well as the co-expression module in which the DDG was identified. The data can be filtered to highlight DDGs that are: a tradeSeq-identified gene for the cell boundary, a gene that was identified by *Al-Barghouthi et al., 2022*, as having eQTL that also colocalizes with BMD GWAS associations, a gene that was identified by *Abood et al., 2023*, as having sQTL that also colocalizes with BMD GWAS associations, a gene that was tested by the IMPC and had a significant effect on BMD when knocked out, or gene that was identified here as an eGene in the cell type-specific eQTL analysis using the 80 DO scRNA-seq data. Other columns of the data contain information at the network level: Number_neighbors=number of nodes (genes) in Bayesian network, Number_tradeSeq_gene_neighbors=number of genes that were also tradeSeq-identified genes for the cell-type boundary, All network neighbors=all genes in network, Neighbors_eQTL_Al_Barghouthi=all genes in the network that were identified from *Al-Barghouthi et al., 2022*, Neighbors_sQTL_Abood=all genes in the network that were identified from *Abood et al., 2023*, Neighbors_IMPC_BMD_gene=all genes in the network that were tested by the IMPC and had a significant effect on BMD when knocked out, Neighbors_DO_eGene=all genes in the network that was identified here as an eGene in the cell type-specific eQTL analysis using the 80 DO scRNA-seq data, Neighbors_tradeSeq_gene_for_boundary=all genes that were also tradeSeq-identified genes for the cell-type boundary.

Supplementary file 3. Supplementary file 3a-g. (a) PANTHER Gene Ontology (GO) Enrichment analysis for differentiation driver genes (DDGs) identified for the late mesenchymal progenitor (LMP) to osteoblast progenitor (OBP) cell-type boundary (osteogenic trajectory 1). (b) PANTHER GO Enrichment analysis for DDGs identified for the OBP to OB1 cell-type boundary (osteogenic trajectory 1). (c) PANTHER GO Enrichment analysis for DDGs identified for the OBP to OB2 cell-type

boundary (osteogenic trajectory 2). (d) PANTHER GO Enrichment analysis for DDGs identified for the LMP to MALP cell-type boundary (adipogenic trajectory). (e) PANTHER GO Enrichment analysis for DDGs identified for the MALP to the end (of the trajectory) cell-type boundary (adipogenic trajectory). (f) Prioritized DDG network analysis for the adipogenic trajectory (26 total, 21 distinct). The enrichment of each prioritized DDG Bayesian network for tradeSeq-identified genes (identified for the cell-type boundary along the associated trajectory) is displayed as nominal and adjusted p-values, as well as the co-expression module in which the DDG was identified. The data can be filtered to highlight DDGs that are: a tradeSeq-identified gene for the cell boundary, a gene that was identified by *Al-Barghouthi et al., 2022*, as having eQTL that also colocalizes with BMD GWAS associations, a gene that was identified by *Abood et al., 2023*, as having sQTL that also colocalizes with BMD GWAS associations, a gene that was tested by the IMPC and had a significant effect on BMD when knocked out, or gene that was identified here as an eGene in the cell type-specific eQTL analysis using the 80 DO scRNA-seq data. Other columns of the data contain information at the network level: Number_neighbors=number of nodes (genes) in Bayesian network, Number_tradeSeq_gene_neighbors=number of genes that were also tradeSeq-identified genes for the cell-type boundary, All network neighbors=all genes in network, Neighbors_eQTL_Al_Barghouthi=all genes in the network that were identified from *Al-Barghouthi et al., 2022*, Neighbors_sQTL_Abood=all genes in the network that were identified from *Abood et al., 2023*, Neighbors_IMPC_BMD_gene=all genes in the network that were tested by the IMPC and had a significant effect on BMD when knocked out, Neighbors_DO_eGene=all genes in the network that was identified here as an eGene in the cell type-specific eQTL analysis using the 80 DO scRNA-seq data, Neighbors_tradeSeq_gene_for_boundary=all genes that were also tradeSeq-identified genes for the cell-type boundary. (g) Expression specificity scores (ESμ) for each gene for each cell cluster of the BMSC-OB scRNA-seq data for the 80 DO mice. ESμ scores are generated during the CELLEX portion of the CELLECT analysis pipeline. ESμ values range from 0 to 1 and are a gene's marginal likelihood of being specifically expressed in a given cell type, where 1 is the most specific and 0 is not specific.

MDAR checklist

## Data availability

The data that support the findings of this study are openly available in NCBI Gene Expression Omnibus database with accession codes GSE152806 and GSE269583. Processed scRNA-seq data available on Zenodo at https://doi.org/10.5281/zenodo.15299630. Code for analysis is available on GitHub at https://github.com/Farber-Lab/DO80_project (copy archived at *Farber, 2025*).

The following datasets were generated:

| Author(s) | Year | Dataset title | Dataset URL | Database and Identifier |
|---|---|---|---|---|
| Dillard LJ, Calabrese GM, Mesner LD, Farber CR | 2025 | Cell type-specific network analysis in Diversity Outbred mice identifies genes potentially responsible for human bone mineral density GWAS associations | https://www.ncbi.nlm.nih.gov/geo/query/acc.cgi?acc=GSE269583 | NCBI Gene Expression Omnibus, GSE269583 |
| Dillard L | 2024 | Processed Seurat Object | https://doi.org/10.5281/zenodo.15299630 | Zenodo, 10.5281/zenodo.15299630 |

The following previously published dataset was used:

| Author(s) | Year | Dataset title | Dataset URL | Database and Identifier |
|---|---|---|---|---|
| Al-Barghouthi B, Mesner L, Calabrese G, Brooks D, Tommasini S, Bouxsein M, Horowitz M, Rosen C, Nguyen K, Haddox S, Farber E, Onengut-Gumuscu S, Pomp D, Farber C | 2020 | Single-cell RNA-seq of bone marrow-derived stromal cells from 5 Diversity Outbred mice | https://www.ncbi.nlm.nih.gov/geo/query/acc.cgi?acc=GSE152806 | NCBI Gene Expression Omnibus, GSE152806 |

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
