## [Editor Report · eLife Assessment]

This is an **important** study that provides **compelling** data from a diverse set of approaches from single cell transcriptome data and network analysis from genetically diverse mouse cells to identify novel driver genes underlying human GWAS associations. The authors present evidence that network analysis of scRNA-seq data from genetically diverse mouse bone-marrow derived stromal cells can be informative for identifying human BMD GWAS driver genes. Their approach should be broadly used and applicable to other GWAS studies.

---

## [Referee Report · Reviewer #1 (Public review)]

In this manuscript, Dillard and colleagues integrate cross-species genomic data with a systems approach to identify potential driver genes underlying human GWAS loci and establish the cell type(s) within which these genes act and potentially drive disease.

Specifically, they utilize a large single cell RNA-seq (scRNA-seq) dataset from an osteogenic cell culture model - bone marrow-derived stromal cells cultured under osteogenic conditions (BMSC-OBs) - from a genetically diverse outbred mouse population called the Diversity Outbred (DO) stock to discover network driver genes that likely underlie human bone mineral density (BMD) GWAS loci. The DO mice segregate over 40M single nucleotide variants, many of which affect gene expression levels, therefore making this an ideal population for systems genetic and co-expression analyses.

The current study builds on previous published work from the same group that used co-expression analysis to identify co-expressed "modules" of genes that were enriched for BMD GWAS associations. In this study, the authors utilized a much larger scRNA-seq dataset from 80 DO BMSC-OBs, inferred co-expression based on Bayesian networks for each identified mesenchymal cell type, focused on networks with dynamic expression trajectories that are most likely driving differentiation of BMSC-OBs, and then prioritized genes ("differentiation driver genes" or DDGs) in these osteogenic differentation networks that had known expression or splicing QTLs (eQTL/sQTLs) in any GTEx tissue that co-localized with human BMD GWAS loci. The systems analysis is impressive, the experimental methods are described in detail, and the experiments appear to be carefully done. The computational analysis of the single cell data is comprehensive and thorough, and the evidence presented in support of the identified DDGs, including Tpx2 and Fgfrl1, is for the most part convincing. Some limitations in the data resources and methods hamper enthusiasm somewhat and are discussed below.

Overall, while this study will no doubt be valuable to the BMD community, the cross-species data integration and analytical framework may be more valuable and generally applicable to the study of other diseases, especially for diseases with robust human GWAS data but for which robust human genomic data in relevant cell types is lacking.

Specific strengths of the study include the large scRNA-seq dataset on BMSC-OBs from 80 DO mice, the clustering analysis to identify specific cell types and sub-types, the comparison of cell type frequencies across the DO mice, and the CELLECT analysis to prioritize cell clusters that are enriched for BMD heritability (Figure 1). The network analysis pipeline outlined in Figure 2 is also a strength, as is the pseudotime trajectory analysis (results in Figure 3).

Potential drawbacks of the authors' approach include their focus on genes that were previously identified as having an eQTL or sQTL in any GTEx tissue. The authors rightly point out that the GTEx database does not contain data for bone tissue, but reason that eQTLs can be shared across many tissues - this assumption is valid for many cis-eQTLs, but it could also exclude many genes as potential DDGs with effects that are specific to bone/osteoblasts. Indeed, the authors show that important BMD driver genes have cell-type specific eQTLs. Another issue concerns potential model overfitting in the iterativeWGCNA analysis of mesenchymal cell type-specific co-expression, which identified an average of 76 co-expression modules per cell cluster (range 26-153). Based on the limited number of genes that are detected as expressed in a given cell due to sparse per cell read depth (400-6200 reads/cell) and drop outs, it's surprising that as many as 153 co-expression modules could be distinguished within any cell cluster. I would suspect some degree of model overfitting is responsible for these results.

Overall, though, these concerns are minor relative to the many strengths of the study design and results. Indeed, I expect the analytical framework employed by the authors here will be valuable to -- and replicated by -- researchers in other disease areas.

Comments on revisions:

Thank you for addressing my concerns. This is an impressive study and manuscript that you should be proud of.

---

## [Referee Report · Reviewer #2 (Public review)]

Summary:

In this manuscript, Farber and colleagues have performed single cell RNAseq analysis on bone marrow derived stem cells from DO Mice. By performing network analysis, they look for driver genes that are associated with bone mineral density GWAS associations. They identify two genes as potential candidates to showcase the utility of this approach.

Strengths:

The study is very thorough and the approach is innovative and exciting. The manuscript contains some interesting data relating to how cell differentiation is occurring and the effects of genetics on this process. The section looking for genes with eQTLs that differ across the differentiation trajectory (Figure 4) was particularly exciting.

Weaknesses:

The manuscript is, in parts, hard to read due to the use of acronyms and there are some questions about data analysis that still need to be addressed.

Comments on revisions:

Dillard et al have made several improvements to their manuscript.

(1) We previously asked the authors to determine whether any cell types were enriched for BMD-related traits since the premise of the paper is that 'many genes impacting BMD do so by influencing osteogenic differentiation or ... adipogenic differentiation'. Given the potential for the cell culture method to skew the cell type distribution non-physiologically, it is important to establish which cell types in their assay are most closely associated with BMD traits. The new CELLECT analysis and Figure 1E address this point nicely. However, it would still be nice to see the correlations between these cell types and BMD traits in the mice as this would provide independent evidence to support their physiological importance more broadly.

(2) Shortening the introduction.

(3) Addressing limitations that arise from not accounting for founder genome SNPs when aligning scRNA-seq data.

(4) The main take-away of this paper is, to us, the development of a single cell approach to studying BMD-related traits. It is encouraging that the cells post-culture appear to be representative of those pre-culture (supplemental figure 3).

However, the authors seem to have neglected several comments made by both reviewers. While we share the authors' enthusiasm for the single cell analytical approach, we do not understand their reluctance to perform further statistical tests. We feel that the following comments have still not been addressed:

(1) The manuscript still contains the following:

"To provide further support that tradeSeq-identified genes are involved in differentiation, we performed a cell type-specific expression quantitative trait locus (eQTL) analysis for each mesenchymal cell type from the 80 DO mice. We identified 563 genes (eGenes) regulated by a significant cis-eQTL in specific cell types of the BMSC-OB scRNA-seq data (Supplementary Table S14). In total, 73 eGenes were also tradeSeq-identified genes in one or more cell type boundaries along their respective trajectories (Supplementary Table S9)."

The purpose of this paragraph is to convince readers that the eGenes approach aligns with the tradeSeq approach (and that their approach can therefore be trusted). It is essential that such claims are supported by statistical reasoning. Given that it would be very simple to perform permutation/enrichment analyses to address this point, and both reviewers requested similar analyses, we do not understand the author's reluctance here. Otherwise, this section should be rewritten so that it does not imply that the identification of these genes provides support for their approach.

(2) Given that a central purpose of this manuscript is to establish a systematic workflow for identifying candidate genes, the manuscript could still benefit from more explanation as to why the authors chose to highlight Tpx2 and Fgfrl1. Tpx2 does already have a role in bone physiology through the IMPC. The authors should comment on why they did not explore Kremen1, for instance, as this gene seems important for the transition to both OB1 and 2.

A final minor comment is that it would be very helpful if the authors could indicate if the DDGs in Table 1 are also eGenes for the relevant cell type. This is much more meaningful than looking through GTEx.

---

## [Author Response]

The following is the authors’ response to the original reviews.

**Reviewer #1 (Public review):**
In this manuscript, Dillard and colleagues integrate cross-species genomic data with a systems approach to identify potential driver genes underlying human GWAS loci and establish the cell type(s) within which these genes act and potentially drive disease. Specifically, they utilize a large single-cell RNA-seq (scRNA-seq) dataset from an osteogenic cell culture model - bone marrow-derived stromal cells cultured under osteogenic conditions (BMSC-OBs) - from a genetically diverse outbred mouse population called the Diversity Outbred (DO) stock to discover network driver genes that likely underlie human bone mineral density (BMD) GWAS loci. The DO mice segregate over 40M single nucleotide variants, many of which affect gene expression levels, therefore making this an ideal population for systems genetic and co-expression analyses. The current study builds on previously published work from the same group that used co-expression analysis to identify co-expressed "modules" of genes that were enriched for BMD GWAS associations. In this study, the authors utilize a much larger scRNA-seq dataset from 80 DO BMSC-OBs, infer co-expression-based and Bayesian networks for each identified mesenchymal cell type, focused on networks with dynamic expression trajectories that are most likely driving differentiation of BMSC-OBs, and then prioritized genes ("differentiation driver genes" or DDGs) in these osteogenic differentiation networks that had known expression or splicing QTLs (eQTL/sQTLs) in any GTEx tissue that colocalized with human BMD GWAS loci. The systems analysis is impressive, the experimental methods are described in detail, and the experiments appear to be carefully done. The computational analysis of the single-cell data is comprehensive and thorough, and the evidence presented in support of the identified DDGs, including Tpx2 and Fgfrl1, is for the most part convincing. Some limitations in the data resources and methods hamper enthusiasm somewhat and are discussed below. Overall, while this study will no doubt be valuable to the BMD community, the cross-species data integration and analytical framework may be more valuable and generally applicable to the study of other diseases, especially for diseases with robust human GWAS data but for which robust human genomic data in relevant cell types is lacking.Specific strengths of the study include the large scRNA-seq dataset on BMSC-OBs from 80 DO mice, the clustering analysis to identify specific cell types and sub-types, the comparison of cell type frequencies across the DO mice, and the CELLECT analysis to prioritize cell clusters that are enriched for BMD heritability (Figure 1). The network analysis pipeline outlined in Figure 2 is also a strength, as is the pseudotime trajectory analysis (results in Figure 3). One weakness involves the focus on genes that were previously identified as having an eQTL or sQTL in any GTEx tissue. The authors rightly point out that the GTEx database does not contain data for bone tissue, but the reason that eQTLs can be shared across many tissues - this assumption is valid for many cis-eQTLs, but it could also exclude many genes as potential DDGs with effects that are specific to bone/osteoblasts. Indeed, the authors show that important BMD driver genes have cell-type-specific eQTLs. Furthermore, the mesenchymal cell type-specific co-expression analysis by iterative WGCNA identified an average of 76 co-expression modules per cell cluster (range 26-153). Based on the limited number of genes that are detected as expressed in a given cell due to sparse per-cell read depth (400-6200 reads/cell) and dropouts, it's hard to believe that as many as 153 co-expression modules could be distinguished within any cell cluster. I would suspect some degree of model overfitting here and would expect that many/most of these identified modules have very few gene members, but the methods list a minimum module size of 20 genes. How do the numbers of modules identified in this study compare to other published scRNA-seq studies that use iterative WGCNA?In the section "Identification of differentiation driver genes (DDGs)", the authors identified 408 significant DDGs and found that 49 (12%) were reported by the International Mouse Knockout [sic] Consortium (IMPC) as having a significant effect on whole-body BMD when knocked out in mice. Is this enrichment significant? E.g., what is the background percentage of IMPC gene knockouts that show an effect on whole-body BMD? Similarly, they found that 21 of the 408 DDGs were genes that have BMD GWAS associations that colocalize with GTEx eQTLs/sQTLs. Given that there are > 1,000 BMD GWAS associations, is this enrichment (21/408) significant? Recommend performing a hypergeometric test to provide statistical context to the reported overlaps here.

We thank the reviewer for their constructive feedback and thoughtful questions. In regards to the iterativeWGCNA, a larger number of modules is sometimes an outcome of the analysis, as reported in the iterativeWGCNA preprint (Greenfest-Allen et al., 2017). While we did not make a comparison to other works leveraging this tool for scRNA-seq, it has been used broadly across other published studies, such as PMID: 39640571, 40075303, 33677398, 33653874. While model overfitting, as you mention, may be a cause for more modules, our Bayesian network analysis we perform after iterativeWGCNA highlights smaller aspects of coexpression modules, as opposed to focusing on the entirety of any given module.

We did not perform enrichment or statistical tests as our goal was to simply highlight attributes or unique features of these genes for additional context.

**Reviewer #2 (Public review):**
Summary:In this manuscript, Farber and colleagues have performed single-cell RNAseq analysis on bone marrow-derived stem cells from DO Mice. By performing network analysis, they look for driver genes that are associated with bone mineral density GWAS associations. They identify two genes as potential candidates to showcase the utility of this approach.Strengths:The study is very thorough and the approach is innovative and exciting. The manuscript contains some interesting data relating to how cell differentiation is occurring and the effects of genetics on this process. The section looking for genes with eQTLs that differ across the differentiation trajectory (Figure 4) was particularly exciting.Weaknesses:The manuscript is in parts hard to read due to the use of acronyms and there are some questions about data analysis that need to be addressed.

We thank the reviewer for their feedback and shared enthusiasm for our work. We tried to minimize the use of technical acronyms as much as we could without compromising readability. Additionally, we addressed questions regarding aspects of data analysis.

**Reviewer #1 (Recommendations for the authors):**
(1) For increased transparency and to allow reproducibility, it would be necessary for the scripts used in the analysis to be shared along with the publication of the preprint. Also, where feasible, sharing the processed data in addition to the raw data would allow the community greater access to the results and be highly beneficial.

Thank you for this suggestion. The raw data will be available via GEO accession codes listed in the data availability statement. We will make available scripts for some analyses on our Github (https://github.com/Farber-Lab/DO80_project) and processed scRNA-seq data in a Seurat object (.rds) on Zenodo (https://zenodo.org/records/15299631)

(2) Lines 55-76: I think the summary of previous work here is too long. I understand that they would like to cover what has been done previously, but this seems like overkill.

Good suggestion. We have streamlined some of the summary of our previous work.

(3) Did the authors try to map QTL for cell-type proportion differences in their BMSC-OBs? While 80 samples certainly limit mapping power, the data shown in Figs 4C/D suggest that you might identify a large-effect modifier of LMP/OB1 proportions.

We did try to map QTL for cell type proportion differences, but no significant associations were identified.

(4) Methods question: Does the read alignment method used in your analysis account for SNPs/indels that segregate among the DO/CC founder strains? If not, the authors may wish to include this in their discussion of study limitations and speculate on how unmapped reads could affect expression results.

The read alignment method we used does not account for SNPs/indels from the DO founder strains that fall in RNA transcripts captured in the scRNA-seq data. We have included this as a limitation in our discussion (line 422-424).

(5) Much of the discussion reads as an overview of the methods, while a discussion of the results and their context to the existing BMD literature is relatively lacking in comparison.

We have added additional explanation of the results and context to the discussion (line 381-382, 396-407).

(6) Figure 1E and lines 146-149: Adjusted p values should be reported in the figure and accompanying text instead of switching between unadjusted and adjusted p values.

We updated Figure 1e to portray adjusted p-values, listed the adjusted p-values in legend of Figure 1e, and listed them in the main text (line 153-154).

(7) Why do the authors bring the IMPC KO gene list into the analysis so late? This seems like a highly relevant data resource (moreso than the GTEx eQTLs/sQTLs) that could have been used much earlier to help identify DDGs.

Given that our scRNA-seq data is also from mice, we did choose to integrate information from the IMPC to highlight supplemental features of genes in networks (i.e., genes that have an experimentally-tested and significant effect on BMD in mice). However, our primary goal was to inform human GWAS and leverage our previous work in which we identified colocalizations between human BMD GWAS and eQTL/sQTL in a human GTEx tissue, which is why this information was used to guide our network analysis.

(8) Does Fgfrl1 and/or Tpx2 have a cis-eQTL in your BMSC-OB scRNA-seq dataset?

We did not identify cis-eQTL effects for Fgfrl1 and Tpx2.

(9) Figure 4B-C: These eQTLs may be real, but based on the diplotype patterns in Figure 4C, I suspect they are artifacts of low mapping power that are driven by rare genotype classes with one or two samples having outlier expression results. For example, if you look at the results in Fig 4C for S100a1 expression, the genotype classes with the highest/lowest expression have lower sample numbers. In the case of Pkm eQTL showing a PWK-low effect, the PWK genome has many SNPs that differ from the reference genome in the 3' UTR of this gene, and I wonder if reads overlapping these SNPs are not aligning correctly (see point 4 above) and resulting (falsely) in lower expression values for samples with a PWK haplotype.

As mentioned above, our alignment method did not consider DO founder genetic variation that is specifically located in the 3’ end of RNA transcripts in the scRNA-seq data. We have included this as a limitation in our discussion (line 422-424).

In future studies, we intend to include larger populations of mice to potentially overcome, as you mention, any artifacts that may be attributable to low statistical power, rare genotype classes, or outlier expression.

**Reviewer #2 (Recommendations for the authors):**
Major Points(1) The authors hypothesize "that many genes impacting BMD do so by influencing osteogenic differentiation or possibly bone marrow adipogenic differentiation". However, cell type itself does not correlate with any bone trait. Does this indicate that the hypothesis is not entirely correct, as genes that drive these phenotypes would not be enriched in one particular cell type? The authors have previously identified "high-priority target genes". So, are there any cell types that are enriched for these target genes? If not, this would indicate that all these genes are more ubiquitously expressed and this is probably why they would have a greater effect on the overall bone traits. Furthermore, are the 73 eGenes (so genes with eQTLs in a particular cell type that change around cell type boundaries) or the DDGs (Table 1) enriched for these high-priority target genes?

The bone traits measured in the DO mice are complex and impacted by many factors, including the differentiation propensity and abundance of certain cell types, both within and outside of bone. Though we did not identify correlations between cell type abundance and the bone traits we measured, we tailored our investigations to focus on cellular differentiation using the scRNA-seq data. However, future studies would need to be performed to investigate any connections between cellular differentiation, cell type abundance, and bone traits.

We did not perform enrichment analyses of either the target genes identified from our other work or eGenes identified here, but instead used the target gene list to center our network analysis and the eGenes to showcase the utility of the DO mouse population.

(2) The readability of the paper could be improved by minimising the use of acronyms and there are several instances of confusing wording throughout the paper. In many cases, this can be solved by re-organising sentences and adding a bit more detail. For example, it was unclear how you arrived at Fgfrl1 or Tpx2.

One of the goals of our study was to identify genes that have (to our knowledge) little to no known connection to BMD. We chose to highlight Fgfrl1 and Tpx2 because there is minimal literature characterizing these genes in the context of bone, which we speak to in the results (line 296-297). Additionally, we prioritized these genes in our previous work and they were identified in this study by using our network analyses using the scRNA-seq data, which we mention in the results (line 276-279).

(3) Technical aspects of the assay. In Figure 1d you show that the cell populations vary considerably between different DO mice. It would be useful to give some sense of the technical variance of this assay given that the assay involves culturing the cells in an exogenous environment. This could take the form of tests between mice within the same inbred strain, or even between different legs of the same DO mice to show that results are technically very consistent. It might also be prudent to identify that this is a potential limitation of the approach as in vitro culturing has the potential to substantially change the cell populations that are present.

We agree that in vitro culturing, in addition to the preparation of single cells for scRNA-seq, are unavoidable sources of technical variation in this study. However, the total number of cells contributed by each of the 80 DO mice after data processing does not appear to be skewed and the distribution appears normal (see added figures, now included as Supplemental Figure 3). Therefore, technical variation is at least consistent across all samples. Nevertheless, we have mentioned the potential for technical variation artifacts in our study in the discussion (line 414-416).

(4) Need for permutation testing. "We identified 563 genes regulated by a significant eQTL in specific cell types. In total, 73 genes with eQTLs were also tradeSeq-identified genes in one or more cell type boundaries". These types of statements are fine but they need to be backed up with permutation testing to show that this level of enrichment is greater than one would expect by chance.

We did not perform enrichment tests as our only goal was to 1. determine if eQTL could be resolved in the DO mouse population using our scRNA-seq data and 2. predict in what cell type the associated eQTL and associated eGene may have an effect.

(5) The main novelty of the paper seems to be that you have used single-cell RNA seq (given that you appear to have already detailed the candidates at the end). I don't think this makes the paper less interesting, but I think you need to reframe the paper more about the approach, and not the specific results. How you landed on these candidates is also not clear. So the paper might be improved by more robustly establishing the workflow and providing guidelines for how studies like this should be conducted in the future.

We sought to not only devise a rigorous approach to analyze our single cell data, but also showcase the utility of the approach in practice by highlighting targets for future research (i.e., Fgfrl1 and Tpx2).

Our goal was to identify novel genes and we landed on these candidate genes (Fgfrl1 and Tpx2) because they had substantial data supporting their causality and they have yet to be fully characterized in the context of bone and BMD (line 295-297).

In regards to establishing the workflow, we have included rationale for specific aspects of our approach throughout the paper. For example, Figure 2 itemizes each step of our network analysis and we explain why each step is utilized throughout various parts results (e.g., lines 168-170, 179-181, 191-193, 202-203, 257-260, 276-277).

We have added a statement advocating for large-scale scRNA-seq from genetically diverse samples and network analyses for future studies (line 436-438).

Minor Points(1) In the summary you use the word "trajectory". Trajectories for what? I assume the transition between cell types, but this is not clear.

We added text to clarify the use of trajectory in the summary (line 34).

(2) This sentence: "By 60 identifying networks enriched for genes implicated in GWAS we predicted putatively causal genes 61 for hundreds of BMD associations based on their membership in enriched modules." is also not clear. Do you mean: we predicted putatively causal genes by identifying clusters of co-expressed genes that were enriched for GWAS genes?" It is not clear how you identify the causal gene in the network. Is this just based on the hub gene?

The aforementioned sentence has since been removed to streamline the introduction, as suggested by Reviewer 1.

In regards to causal gene identification, it is not based on whether it is hub gene. We prioritized a DDG (and their associated networks) if it was a causal gene that we identified in our previous work as having eQTL/sQTL in a GTEx tissue that colocalizes with human BMD GWAS.

(3) Figure 3C. This is good but the labels are quite small. Would be good to make all the font sizes larger.

We have enlarged Figure 3C.

(4) Line 341 in the Discussion should be "pseudotemporal".

We have edited “temporal” to “pseduotemporal”.